# Quality Assurance Framework for Recovered Binders and Aggregates from Asphalt Mixtures Incorporating Recycled Materials

Eslam Deef-Allah [1,2] and Magdy Abdelrahman [1,*]

[1] Department of Civil, Architectural and Environmental Engineering, Missouri University of Science and Technology, Rolla, MO 65409, USA; emddkc@mst.edu

[2] Department of Construction Engineering and Utilities, Faculty of Engineering, Zagazig University, Zagazig 44519, Egypt

[*] Correspondence: m.abdelrahman@mst.edu

**Abstract:** This study proposes that a proactive quality assurance (QA) framework for asphalt mixes with recycled materials, i.e., reclaimed asphalt pavement and recycled asphalt shingles, should be developed. Quality control (QC) is generally concerned with the contractor's obligation to produce mixes which meet the job mix formula (JMF) targets. However, QA considers the variability in fabrication processes and materials and offers monitoring by evaluating the contractor's performance. Although both aggregate gradations and asphalt contents were within the JMF specifications, the recovered binders revealed significant differences from the contract binders in the JMF. Rheological tests indicated increased stiffness and elasticity but reduced capability to relax thermal stresses in binders recovered from plant–lab- and lab-fabricated mixtures, compared to field mixtures. Thermal-rheological analysis models corroborated these results by demonstrating reduced decomposition areas for more aged binders, enhancing performance prediction—especially for limited binder amounts. The creation of a QA decision matrix facilitated uniform, performance-oriented assessments. The matrix indicated only 23% of the mixtures satisfied JMF criteria and reported QC data—predominantly field mixtures—underscoring the impact of the fabrication mechanisms and the use of soft binders. This matrix integrates statistical analysis and binder performance assessments as a tool for verifying material compliance and tracking contractor efficiency. It reflects a transition from traditional QC toward a more proactive QA framework for sustainable pavements.

**Keywords:** quality assurance; RAP; RAS; decision matrix; interactions; fabrication mechanism; recovered binders; ANOVA

## 1. Introduction

Quality control (QC) and quality assurance (QA) in the pavement industry are the paramount factors in fabricating pavements that perform and last with fewer defects [1,2]. Modern asphalt mixtures that include recycled materials rely heavily on effective QA/QC procedures because their material properties exhibit greater variation. QC pertains to the contractor's direction and oversight of the asphalt characteristics to guarantee that the job mix formula (JMF) requirements are met [3,4]. Conversely, QA, overseen by owners and transportation companies (including departments of transportation), entails systematic oversight and verification of the contractor's QC procedures to ensure adherence to contractual and regulatory standards [5,6]. Ensuring pavement integrity while effectively

incorporating recycled materials relies on a robust QA system, in light of the increasing emphasis on sustainability.

Numerous studies have clarified the difficulties associated with the incorporation of reclaimed asphalt pavement (RAP) and recycled asphalt shingles (RASs) in asphalt mixtures [7–9]. RAP refers to materials extracted and processed from the pavement, encompassing worthwhile components such as aggregate and asphalt binder [10–12]. Conversely, RASs are materials derived from the processing of waste asphalt shingles, which may originate from manufacturing surplus or post-consumer roofing removals. These RASs contain asphalt binders, polymers, fibers, fillers, and aggregates [13,14]. The aged components of binders in RAP and oxidized air-blown binders in RASs modify the performance of total recovered binders. Moreover, the properties of binders in the RAP and RASs are different from one source to another [12,15,16]. From the QC/QA standpoint, additional studies focused on the importance of the asphalt contents (ACs) and aggregates' skeletons on the mechanical properties and durability values of asphalt mixtures [17–21]. Given how aggregate gradations, binder properties, and recycled components could affect pavement performance [22–24]. It is essential to discern these characteristics through the application of suitable techniques, such as binder extraction and recovery. The centrifuge extraction and rotary evaporator recovery of asphalt binders are common practices in asphalt research and QC/QA laboratories [25–27]. These methodologies are particularly effective with recycled components, as there is little change in the inherent properties of the binder in processing [28–30]. The determination of aggregate gradation is also an important factor due to its effects on stability and performance in terms of mixture compaction, permeability, and the degree of resistance to deformation [31,32].

Given that the properties of recycled materials are becoming more variable, previous studies focused on extracting and recovering binders and aggregates and evaluating the binders' characteristics [30,33–36]. Different aspects were covered in these studies from the QC perspective, including physical, rheological, and chemical analysis of the extracted binders, as well as assessments of the asphalt contents and aggregate gradations. Some research efforts also examined for mixtures incorporating RAP-RAS. However, significant gaps remained, particularly in capturing the broader implications of binder aging, variability in recycled materials, and the interactions between these binders and the original ones in different fabricated mixtures (field, plant, and lab) with the same components from a QA standpoint. The interactions between the aged binders in recycled components and original binders were the primary control in influencing the performance of the total recovered binders [16]. These interactions were reflected in the various fabrication mechanisms of asphalt mixtures (field, plant, and lab), which had not been investigated from the QA standpoint in previous studies. Besides these gaps, another challenge was focusing on predicting the performance of limited amounts of binders via traditional rheological techniques.

Such gaps and challenges, however, showed that it is necessary to have a stricter and more flexible QA framework for asphalt mixtures, including recycled materials, as a result of guaranteeing optimal pavement performance. Contributions to the research included the establishment of a solid QA framework that guarantees alignment not only among the asphalt and aggregate components but also uncovers the interactions between original and recycled binders in mixtures produced from different fabrication mechanisms (field, plant, and lab). Research also combined the thermal studies with the rheological tests to accomplish a more realistic and integrated binder performance prediction. Another tangible contribution of this research was the development of models correlating rheological and thermal properties, enhancing binder performance comprehension. By incorporating the QA framework, the study not only strengthens the QA framework but also provides road authorities with a QA decision matrix to monitor and optimize the use of recycled

components, track contractor utilization of recycled materials, and ensure compliance with performance and sustainability targets in asphalt production.

## 2. Results and Discussion

### 2.1. Gradation Consistency of Extracted Aggregates

Sustaining a consistent aggregate gradation by production sources is paramount to quality, having a direct impact on mixture uniformity, performance, and long-term durability while providing production QC/QA checks for minimizing variability and sustaining integrity. The gradation curves of aggregates extracted from field (F) and plant–lab-compacted (PL) M1 to M4 mixtures were thus compared and analyzed (see Figure 1). In addition, the figure reveals the gradation of F, PL, lab (L), and JMF aggregates for the M2 and M4 mixtures. All aggregates exhibited a 100% passing percentage on the sieves ranging from 2 inches to 3/4 inch, which was omitted from Figure 1. For sieve sizes greater than 3/8 inches, the aggregate gradations were highly consistent, exhibiting that the mixture production and binder extraction procedures had no significant influence on aggregate gradation. The observations confirmed that the intended aggregate gradations in the JMF were closely adhered to while fabricating the various mixtures (e.g., F, PL, and L). However, slight discrepancies were noticed between sieve No. 4 and sieve No. 200, particularly for M1 and M4 aggregates. Therefore, statistical analyses were conducted to compare the results using an analysis of variance (ANOVA) (note Table 1). The *p*-values (Prob > F) varied from $8.20 \times 10^{-1}$ to $9.99 \times 10^{-1}$, greater than $\alpha = 0.05$, indicating no significant difference between the aggregate gradations. In practice, F-ratios close to zero confirm that any detected gradation differences are more likely attributable to random fluctuation than systematic differences produced by the source (F, PL, or L).

**Table 1.** Aggregates gradations ANOVA results.

| Mixture | Source | DF [a] | SS [b] | MS [c] | F Ratio | Prob > F |
|---|---|---|---|---|---|---|
| M1 | Gradation (F and PL) | 1 | 87.81 | 87.80 | $5.28 \times 10^{-2}$ | $8.20 \times 10^{-1}$ |
| | Error | 24 | 39,936.25 | 1664.01 | | |
| | C. Total | 25 | 40,024.06 | | | |
| M2 | Gradation (F, PL, L, and JMF) | 3 | 23.29 | 7.76 | $4.80 \times 10^{-3}$ | $9.99 \times 10^{-1}$ |
| | Error | 48 | 77,574.76 | 1616.14 | | |
| | C. Total | 51 | 77,598.05 | | | |
| M3 | Gradation (F and PL) | 1 | 0.56 | 0.56 | $3.00 \times 10^{-4}$ | $9.85 \times 10^{-1}$ |
| | Error | 24 | 38,746.22 | 1614.43 | | |
| | C. Total | 25 | 38,746.78 | | | |
| M4 | Gradation (F, PL, L, and JMF) | 3 | 57.34 | 19.11 | $1.13 \times 10^{-2}$ | $9.98 \times 10^{-1}$ |
| | Error | 48 | 81,337.69 | 1694.54 | | |
| | C. Total | 51 | 81,395.03 | | | |
| M5 | Gradation (F and JMF) | 1 | 2.20 | 2.20 | $1.40 \times 10^{-3}$ | $9.70 \times 10^{-1}$ |
| | Error | 24 | 37,385.67 | 1557.74 | | |
| | C. Total | 25 | 37,387.87 | | | |
| M6 | Gradation (F and JMF) | 1 | $3.63 \times 10^{-3}$ | $3.63 \times 10^{-3}$ | $2.32 \times 10^{-6}$ | $9.99 \times 10^{-1}$ |
| | Error | 24 | 37,577.53 | 1565.73 | | |
| | C. Total | 25 | 375,577.54 | | | |

**Table 1.** *Cont.*

| Mixture | Source | DF [a] | SS [b] | MS [c] | F Ratio | Prob > F |
|---|---|---|---|---|---|---|
| M7 | Gradation (F and JMF) | 1 | 0.03 | 0.03 | $1.85 \times 10^{-5}$ | $9.96 \times 10^{-1}$ |
| | Error | 24 | 38,373.55 | 1598.90 | | |
| | C. Total | 25 | 38,373.58 | | | |

[a] Degrees of freedom, [b] sum of squares, and [c] mean square.

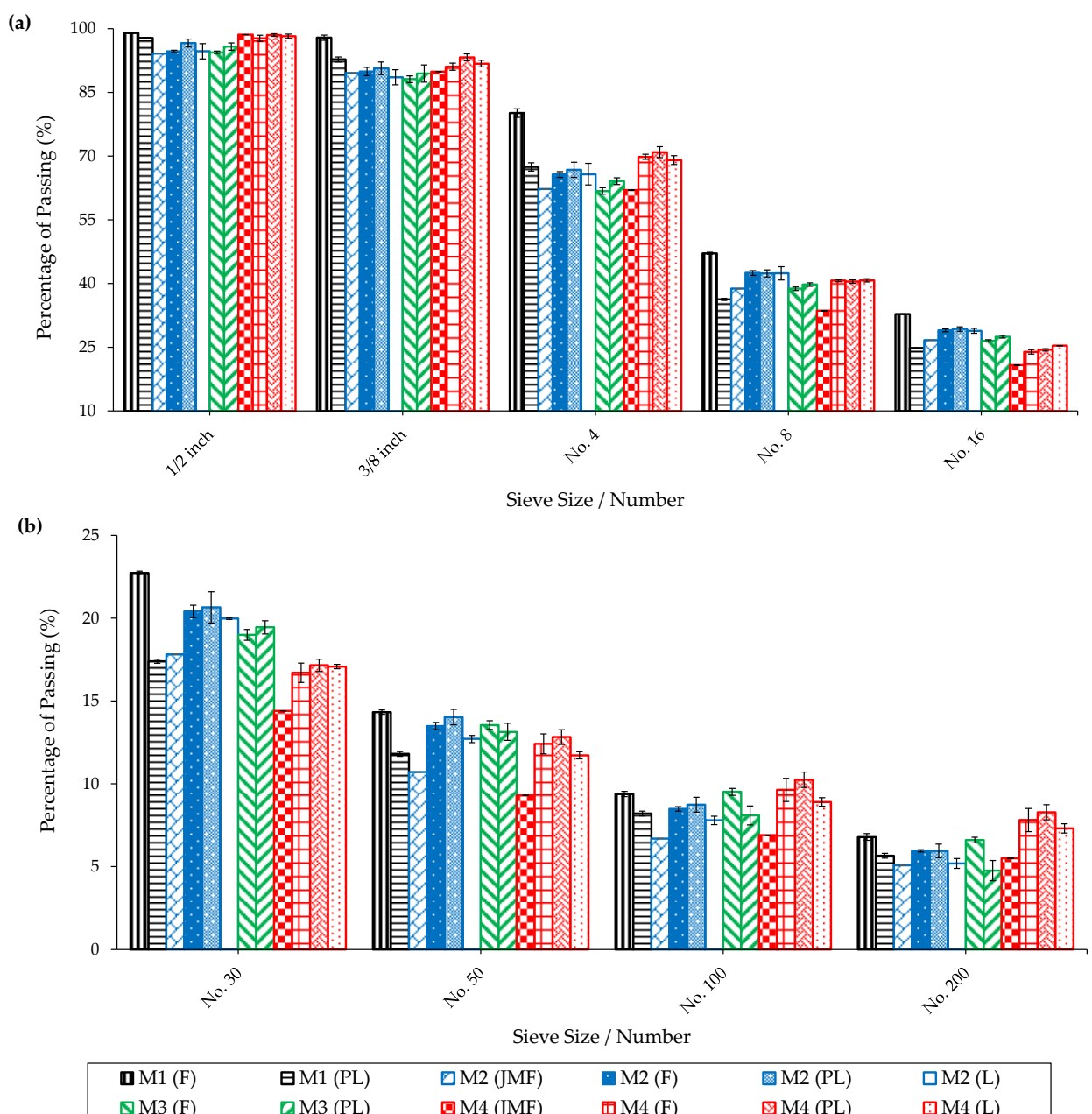

**Figure 1.** Comparisons of extracted aggregate gradations from M1 to M4 mixtures (**a**) No. 16 and larger sieves; (**b**) sieves smaller than No. 16.

### 2.2. Binder Quantitative Analyses

The percentage of AC in asphalt mixtures plays a pivotal role in controlling the performance of the asphalt mixtures. To further understand the impacts of extraction processes on the extracted AC percentages, the contract via extracted AC percentages was compared for the mixtures with no recycled materials [37]. A *p*-value of 0.69 indicated

that there was no significant difference between the extracted and contract AC percentages. However, the use of recycled materials increased the extraction time due to the aged binders in the recycled components, which may have an impact on the extracted AC percentages. Thus, in this section, the QA framework included the assurance of the extracted AC percentages from various fabricated mixtures with the same components. The AC included contract (C) and extracted (E) percentages determined via ashing [EF (ash.), EPL (ash.), and EL (ash.)], centrifuge [EF (centr.), EPL (centr.), and EL (centr.)], and both methods [EF (tot.), EPL (tot.), and EL (tot.)] for the field (EF), plant-and-lab-compacted (EPL), and lab (EL) mixtures. Figure 2 shows comparisons between the contract and extracted AC percentages. For 100% of the samples, the EPL and EL AC percentages exhibited marginally higher values than the EF AC percentages. This might be the reason for the different interaction processes that occurred between the RAP-RAS and the original binders [16]. Hence, the PL and L mixtures showed higher levels of interactions than F mixtures. Moreover, in all samples, the percentages of AC centrifuge were superior to those of AC ashing. This demonstrates the precision of the centrifuge method for determining mineral content compared to the ashing technique.

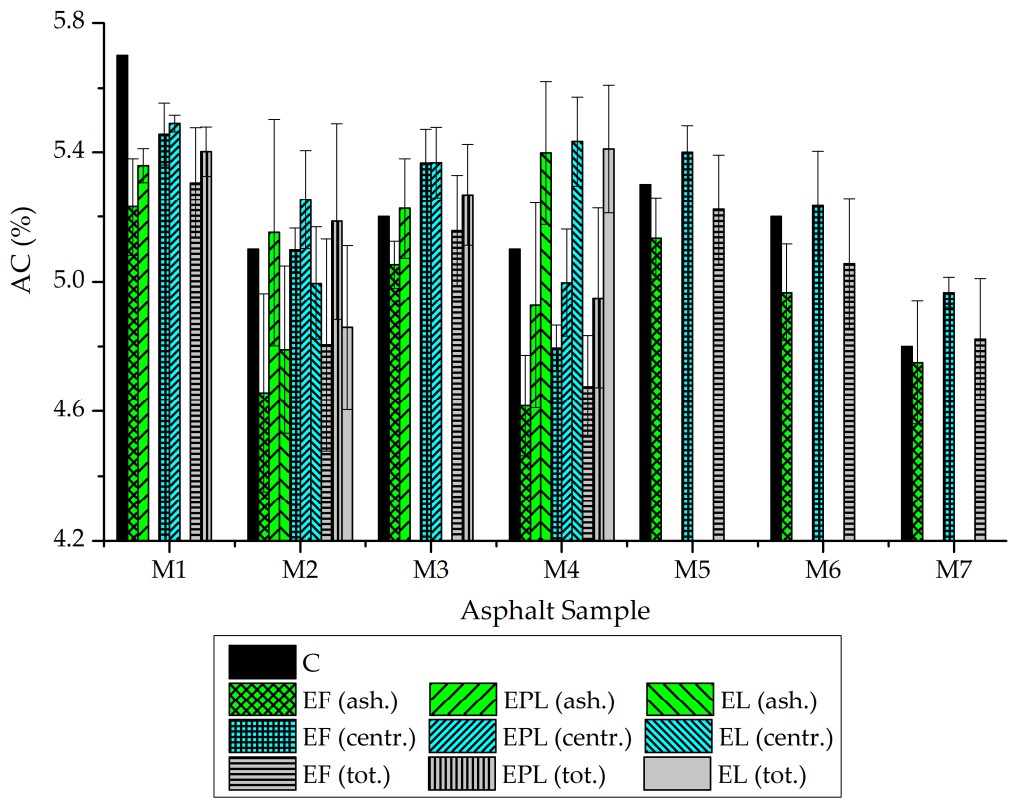

**Figure 2.** Comparisons of contract and extracted AC percentages.

However, an ANOVA was conducted to ascertain the impact of the fabrication mechanism and mineral matter determination on the total extracted AC percentages when compared to the contract AC percentages. Table 2 presents the effect of the fabrication method on the extracted AC percentages by ANOVA. The C AC and PL AC percentages had the highest means (5.20), followed by the mean of the L AC percentages (5.13), and the mean of the F AC percentages (5.00). A *p*-value greater than the significance level (0.05) determined no statistically significant differences between the extracted AC percentages from F, PL, and L mixtures when compared to the C AC percentages. Further, with an F-ratio value less than 1, the observed AC differences between groups could be a random error instead of a meaningful difference between groups.

**Table 2.** Fabrication method effects on extracted AC percentages via ANOVA.

| Source | DF | SS | MS | F Ratio | Prob > F |
|---|---|---|---|---|---|
| AC% (F, PL, L, and C) | 3 | 0.16 | 0.05 | 0.82 | 0.50 |
| Error | 16 | 1.05 | 0.07 | | |
| C. Total | 19 | 1.21 | | | |

The effect of the mineral matter determination methods (ashing and centrifuge) on the extracted AC percentages was compared to the C AC percentages, and the results are presented in Table 3. The F-ratio was 2.72 (greater than 1), indicating variability between AC% C, AC% E (ash.), and AC% E (centr.). Nevertheless, the *p*-value was greater than 0.05, indicating no significant difference between the groups. Pairwise comparisons were performed using statistical tests, which showed no significant differences between the groups (*p*-values > 0.05). The wide confidence intervals, as well as the fact that they all included zero, supported that the differences in AC percentages for (C, ash, centr.) were not statistically significant. The analysis for extracted AC versus the contract AC percentages, concerning the method of fabrication (F, PL, and L) and also whether the mineral matter was determined employing ashing or centrifuge, was carried out using an ANOVA, as presented in Table 4. Since the *p*-value was >0.05, it can be concluded that the factors investigated by the method of fabrication and the determination of mineral matter during the binder extraction process had no statistically significant effects on the extracted AC percentages versus the contract AC percentages. The F-ratio was 1.08; however, pairwise comparisons showed that there were no significant differences between groups through *p*-values > 0.05 and wide confidence intervals including zero.

**Table 3.** Mineral matter determination effects on extracted AC percentages via ANOVA and pairwise comparisons.

| 1. ANOVA Results | | | | | |
|---|---|---|---|---|---|
| Source | DF | SS | MS | F Ratio | Prob > F |
| AC% (ash., centr., and C) | 2 | 0.32 | 0.16 | 2.72 | 0.08 |
| Error | 36 | 2.14 | 0.06 | | |
| C. Total | 38 | 2.47 | | | |
| **2. Pairwise Comparisons** | | | | | |
| Group 1 | Group 2 | Difference | Std Error | t Ratio | Prob > \|t\| | Lower 95% | Upper 95% |
| AC% (ash.) | AC% (centr.) | −0.20 | 0.11 | −2.08 | 0.11 | −0.43 | 0.04 |
| AC% (ash.) | AC% C | −0.19 | 0.11 | −1.95 | 0.14 | −0.42 | 0.05 |
| AC% (centr.) | AC% C | 0.01 | 0.11 | 0.13 | 0.99 | −0.22 | 0.25 |

**Table 4.** Overall AC percentages via ANOVA.

| Source | DF | SS | MS | F Ratio | Prob > F |
|---|---|---|---|---|---|
| AC% (AC% categories listed in Figure 2) | 9 | 0.61 | 0.07 | 1.08 | 0.40 |
| Error | 36 | 2.27 | 0.06 | | |
| C. Total | 45 | 2.88 | | | |

### 2.3. Binder Qualitative Analyses

In this section, comparisons were conducted between the rheological properties of the original and recovered binders at high, intermediate, and low temperatures. These comparisons help elucidate the interactions within the various fabricated mixtures, introducing

reasons for the variations in acceptance levels observed in the proposed QA decision matrix. Furthermore, they help interpret the variability introduced by the recycled components and their effect on the overall properties of the recovered binders.

### 2.3.1. High-Temperature Rheological Analyses

In mixtures without recycled materials [37], recovered binders had 17% lower average rutting parameters ($G^*/\sin\delta$) at 64 °C compared to the original binders. These findings do not indicate any issues with the stiffness of the recycled binders when compared to the originals; nevertheless, the situation may be considerably different for mixtures involving recycled materials. Consequently, the high-temperature rheological analyses of both the recovered and original binders are compared in this section. These analyses are represented in the $G^*/\sin\delta$ at 64 °C in Figure 3, the percentage change in $G^*/\sin\delta$ between the recovered and original binders in Figure 4, and the average high-performance grade (PG) temperature in Figure 5. The results indicated that the high-temperature performance of the recovered binders was determined by other factors aside from the percentages of recycled materials and original binder grades of the asphalt mixtures. The production process itself was also a significant influencing factor. For instance, the M1 mixtures had the lowest amount of recycled material [17% asphalt binder replacement (ABR) from RAP] and a stiff original binder ($G^*/\sin\delta$ = 5.18 kPa). Additionally, with $G^*/\sin\delta$ values below 1.30 kPa, mixtures M2 through M4, with ABR%s ranging from 31% to 35%, displayed less stiff original binders. Nevertheless, the percentage changes in $G^*/\sin\delta$ for the recovered binders were significantly improved by raising the ABR% from 17% to 35% and changing the fabrication mechanism from F to PL or L.

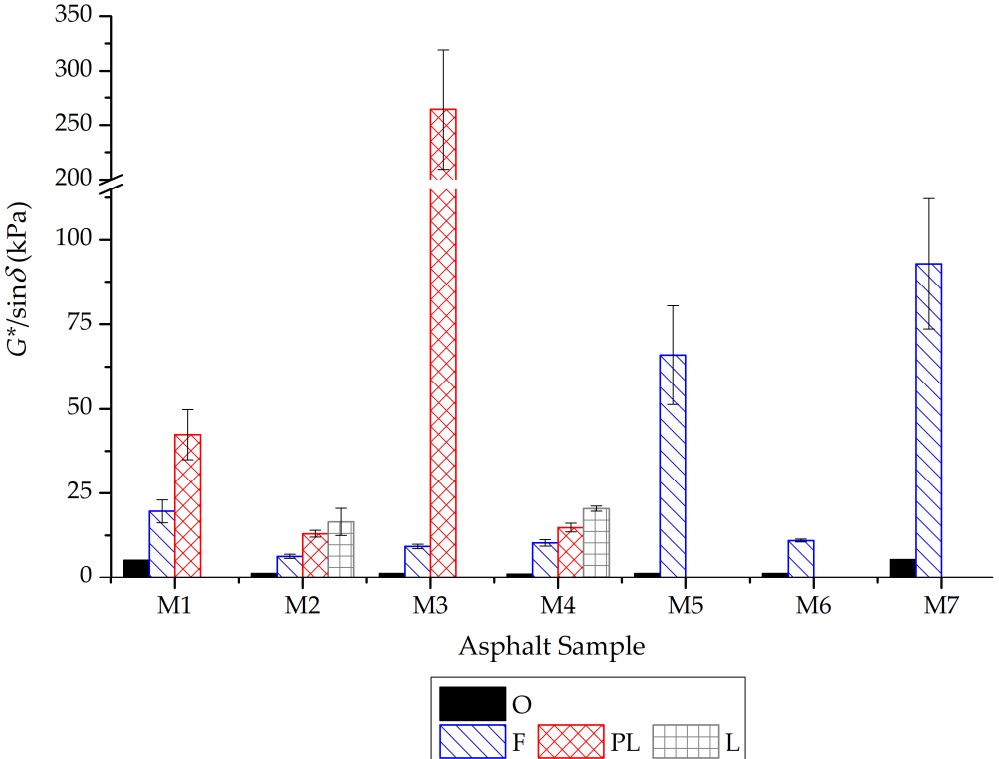

**Figure 3.** Rutting parameters for the original and recovered binders.

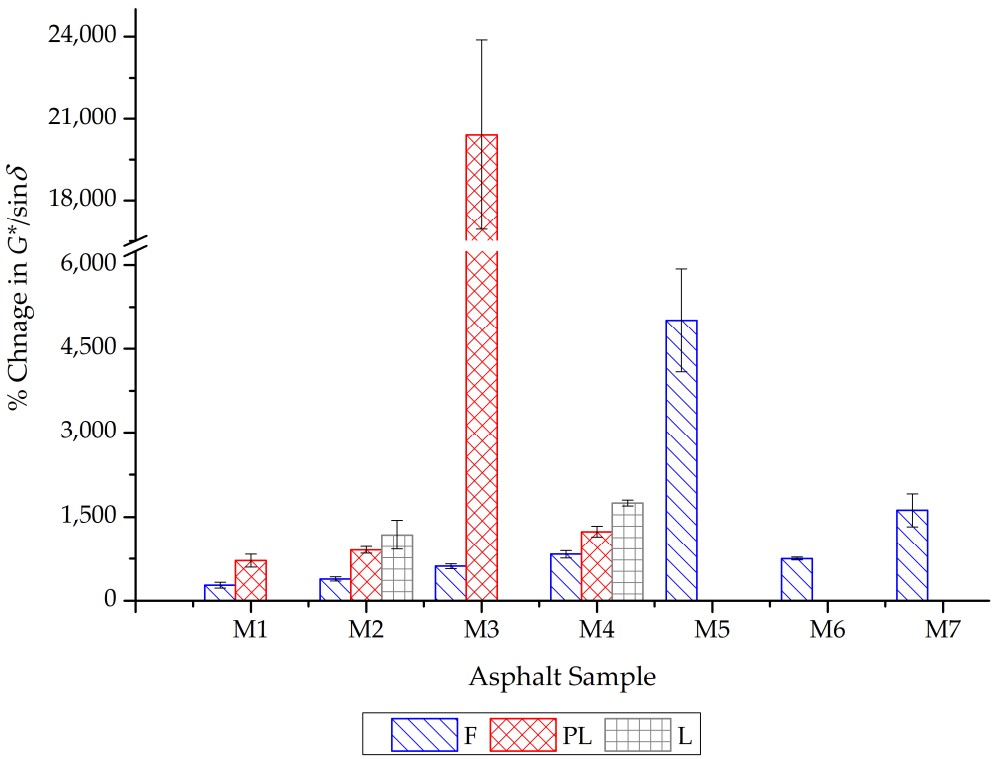

**Figure 4.** Percentage of change in the rutting parameter of the recovered binders.

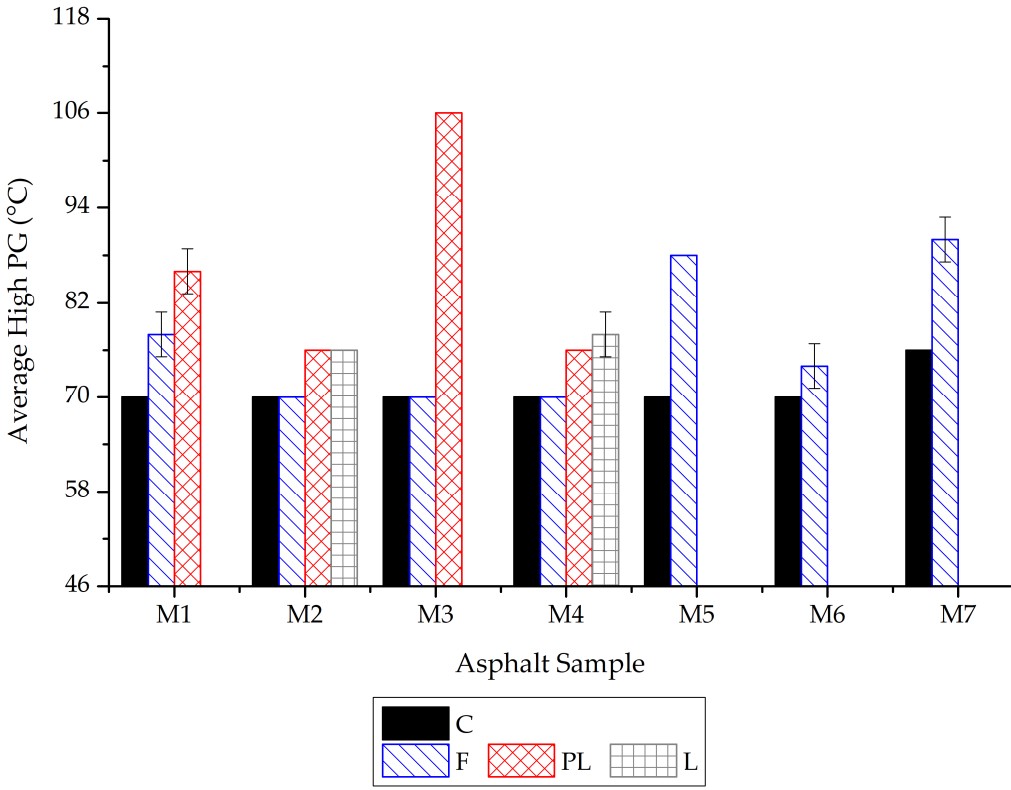

**Figure 5.** Average high PGs for the contract and recovered binders.

The M3, M5, and M6 field mixtures utilized an identical original binder with a $G^*/\sin\delta$ of 1.29 kPa. The M3, M5, and M6 included 33% ABR; however, their recovered binders showed significant differences in the $G^*/\sin\delta$ values. The M5-F recovered binder exhibited a $G^*/\sin\delta$ increase of 613% compared to the M3-F binder and 499% compared to the M6-F binder. The M5 had 33% ABR by RAP, the M3 had 33% ABR by RAS, and the M6 had

18–15% ABR by RAP-RAS. RAP binders interacted more readily with the original binders than RAS binders, causing higher stiffness values. Nonetheless, increased interactions transpired in the PL mixtures, enhancing the interactions between RAS and the original binders (e.g., M3-PL). The M7-F recovered binder was the stiffest among M2-F, M4-F, and M5-F recovered binders, with a comparable percentage of ABR% by RAP. This was related to the lower AC% in the M7, the stiffest original binder, and the variability in the RAP properties between the different sources. From Figure 4, the percentage change in the $G^*/\sin\delta$ range for F binders was from 278.80% to 5011.011%, for PL binders was from 717.94% to 20,407.57%, and for L binders was from 1181.80% to 1746.00%.

Figure 5 illustrates that the contract high PGs were inferior to the high PGs of the recovered binders, highlighting the impact of RAP-RAS binders on the stiffness of the overall recovered binders. The changing of the way mixtures were prepared from the F to the PL or the L also increased the average high PG temperature of the binders. M2 and M3 mixtures were prepared from the same original binder. The only difference between the M3 mixture and the M2 mixture is that the former contains RAS rather than RAP. The M3 asphalt mix includes 33% ABR with RAS instead of 31% ABR with RAP, which is the main difference between it and the M2 mixture. Unlike the oxidized "air-blown" binder in RAS, the aged binder in RAP interacts differently with the original binder [10,38]. The PL mixtures showed greater interactions than the F mixtures, reflecting this finding. The percentage change in the $G^*/\sin\delta$ in the M2 recovered binders was 2.33 times higher for the PL mixtures than the F mixtures. Nevertheless, this percentage change was 33.07 times higher for the M3 PL-recovered binder than the F-recovered binder. The findings were further substantiated by the average high PG temperature in Figure 5: The M3 recovered binder average high PG temperature altered from 70 °C (F) to 106 °C (PL). Additionally, the results shed light on the greater aged behavior variability of the RAS binder compared to the RAP binder, as evidenced by the standard deviation values in Figures 3 and 4, which show that M3 PL-recovered binders had the highest standard deviation values compared to the other recovered binders. This finding suggests that RAS binders experience more severe aging and also a less stable degradation pattern; thus, the most common cause of these phenomena is the lightning of the initial oxidation levels and the effectiveness of the mixing of the aged RAS binder with the original binder. In general, PL- and L-recovered binders showed higher stiffnesses and elasticities than F-recovered binders, which indicates that the different fabrication techniques might have been the reason that the original and recycled binders would interact.

### 2.3.2. Intermediate-Temperature Rheological Analyses

This section is devoted to an examination of the intermediate-temperature rheological properties of original and recovered binders. The fatigue parameters ($G^*.\sin\delta$) at 22 °C versus $G^*/\sin\delta$ at 64 °C is represented in Figure 6. As demonstrated in the figure, the correlation between $G^*/\sin\delta$ and $G^*.\sin\delta$ unveils four discernible zones based on the $G^*/\sin\delta$ = 2.2 kPa and $G^*.\sin\delta$ = 5000 kPa thresholds. Regarding the recovered binders, 23% were located in the lower-right zone of the graph; binders in this zone could resist rutting at 64 °C and fatigue cracking at 22 °C. However, 77% of the recovered binders were situated in the upper-right zone, where they possessed the ability to resist rutting at 64 °C but failed to resist fatigue cracking at 22 °C.

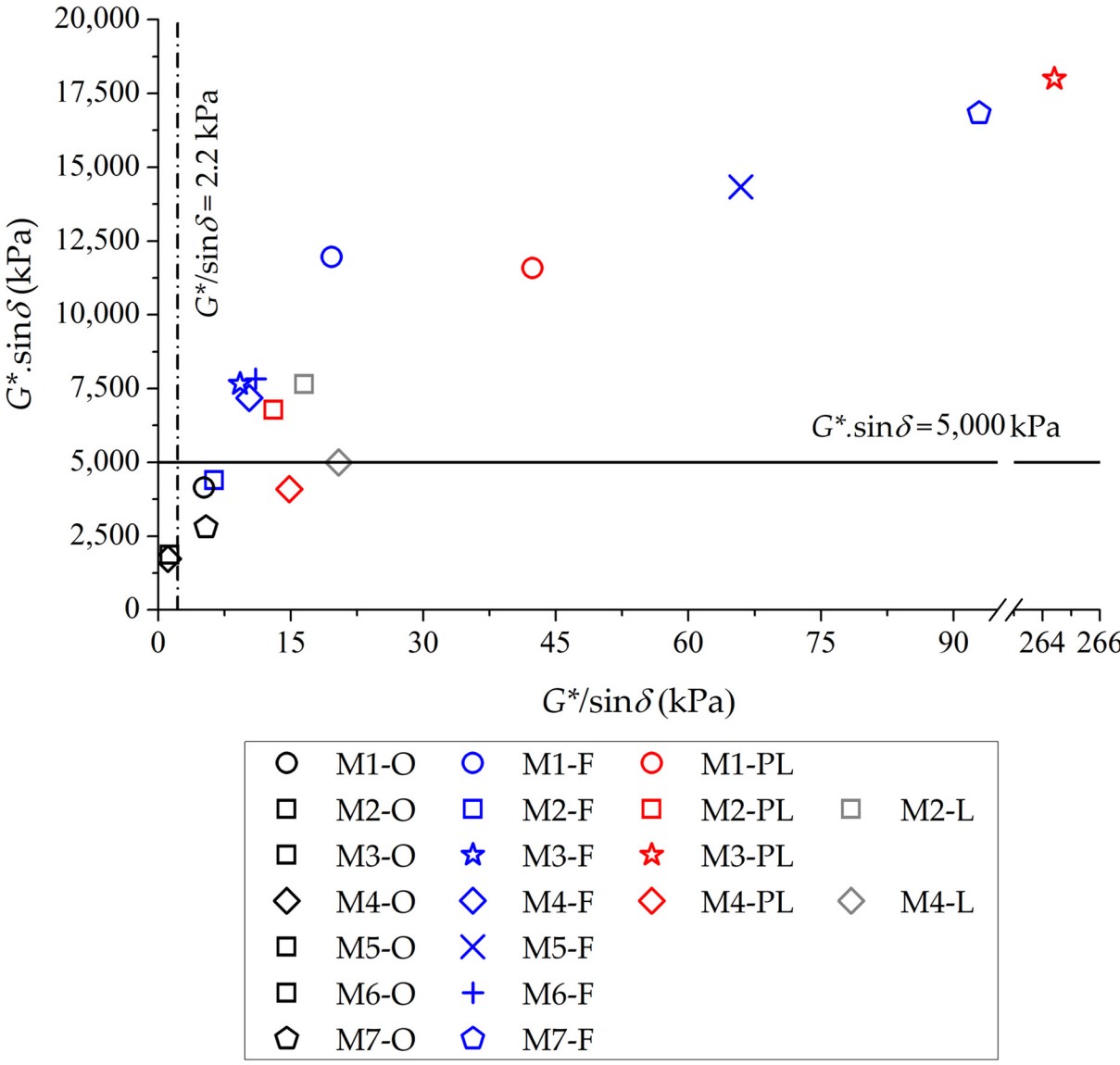

**Figure 6.** Fatigue versus rutting parameter for original and recovered binders.

The Glover–Rowe (G-R) parameter at 15 °C is illustrated in Figure 7, which shows the ductility changes in binders due to aging. It has been observed that two G-R limits have been proposed based on the G-R values, 180 kPa and 600 kPa [10,39]. Block cracking is not signaled by G-R values below 180 kPa; however, this lower threshold is the point of deterioration or initial raveling. Surface cracking makes the deterioration more apparent on the upper side of 600 kPa, while G-R values above this level govern block cracking. The original binders exhibited commendable resistance to block cracking, demonstrating G-R values below 180 kPa. For the recovered binders, 23% of the binders had G-R values lower than 180 kPa, and 46% of the binders showed G-R values between 180 and 600 kPa (damage zone). However, 31% of the recovered binders had G-R values greater than 600 kPa. Furthermore, increasing the interaction processes between the RAP-RAS binders caused more stiffness by increasing the recycled binder's contribution to the total recovered binders. This was concluded because the G-R values of the L and PL binders were higher than those obtained for the F binders. Consequently, incorporating recycled materials in asphalt mixtures improves the recovered binders' stiffness (higher $G^*$ values) and elasticity (lower $\delta$ values). Such material improvement decreases the resistance to fatigue and

block cracking, as manifested by the increase in $G^*.\sin\delta$ and G-R values presented in Figures 6 and 7, respectively.

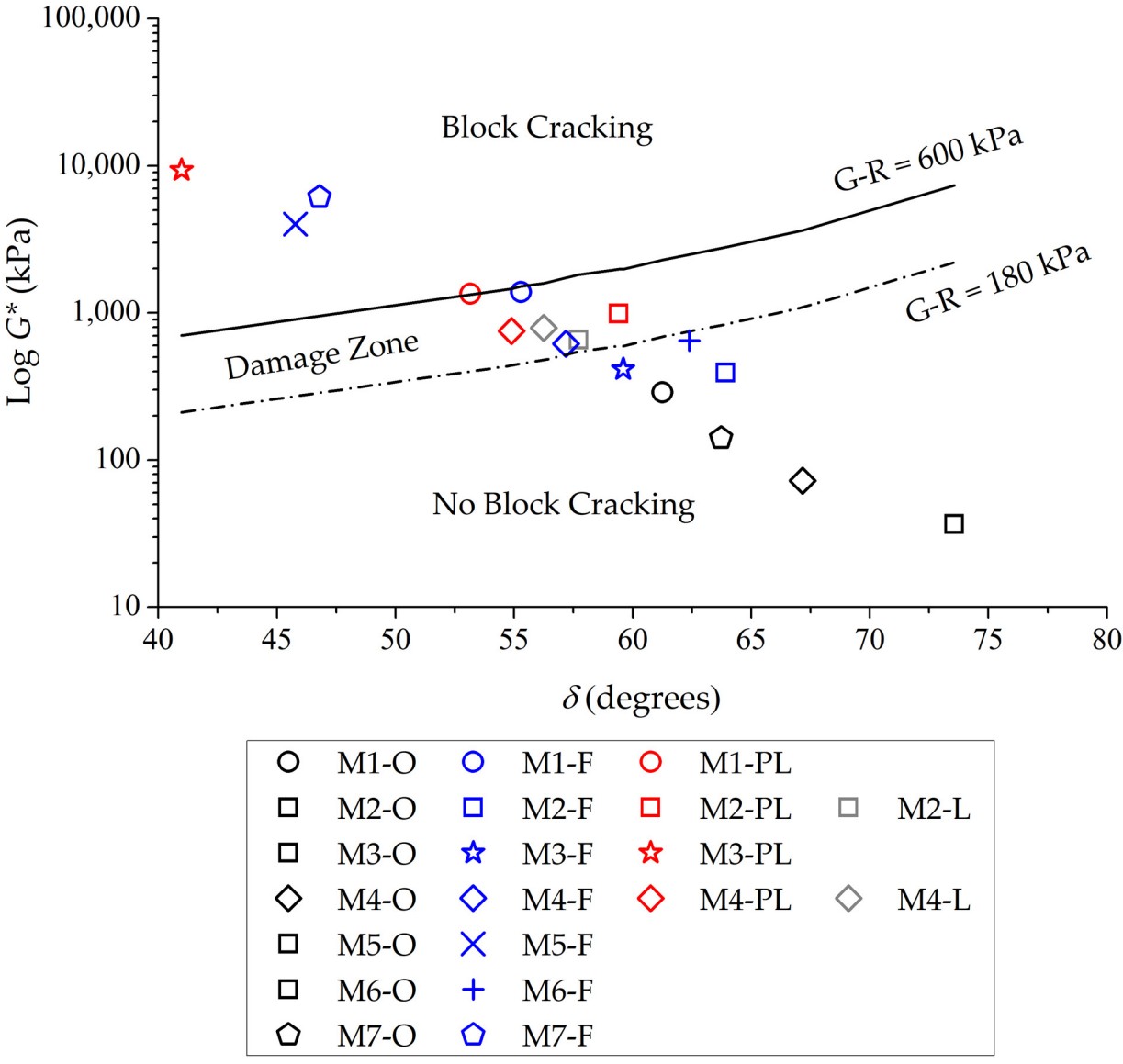

**Figure 7.** G-R parameter for original and recovered binders.

The M3, M5, and M6, utilizing the same original binder, exhibited varying resistances to intermediate temperatures. The recovered binder from the M5-F with 33% ABR by RAP exhibited higher $G^*.\sin\delta$ and G-R values compared to the values of the recovered binders from the M3-F with 33% ABR by RAS and the M6-F with 18–15% ABR by RAP-RAS. This occurred due to the RAP and original binders' rapid interactions compared to the RAS and original binders' interactions, which were elucidated in the high-temperature performance results. The M7-F binder has the worst resistance to fatigue and block cracking among the recovered binders from field mixtures. This was related to the lower AC% in the M7 and the variability in the RAP properties between the different sources. The increasing ABR% invariably leads to a trade-off between enhanced rutting resistance at elevated temperatures and significantly diminished resistance to fatigue and block cracking at intermediate temperatures, particularly in mixtures incorporating RAS. Overall, PL and F mechanisms consistently yield higher values of $G^*.\sin\delta$ and G-R than the original binders, and the results from the L-fabricated samples frequently exhibit intermediate behavior.

Figures 8 and 9 show the percentage change in the $G^*.\sin\delta$ and G-R parameters, respectively. The $G^*.\sin\delta$ percentage change for M4-L was 189.26%, and it varied from 136.80% for M4-PL to 315.68% for M4-F. The pattern followed by the G-R parameter was also similar: The percentage change in the G-R for M4-L was 2402.62%, with 2495.68% for M4-PL, and 1697.37% for M4-F. A similar observation was made for the M2 G-R values, with the percentage change in the G-R for the M2-L recorded at 7147.93%, ranging from 9897.11% for the M2-PL to 2711.22% for the M2-F. The M4-L recovered binders exhibited higher resistance to rutting at elevated temperatures when compared to the PL- and F-recovered binders. However, the M4-L recovered binders also introduced higher resistance to block cracking than the M4-PL recovered binders and to fatigue cracking than the M4-F recovered binders at intermediate temperatures. Completion is necessary for an assessment of the entire performance scale, from high to intermediate and low-temperature behavior, to arrive at a balanced and durable asphalt mix, particularly when recycled materials are included, particularly from the QA standpoint.

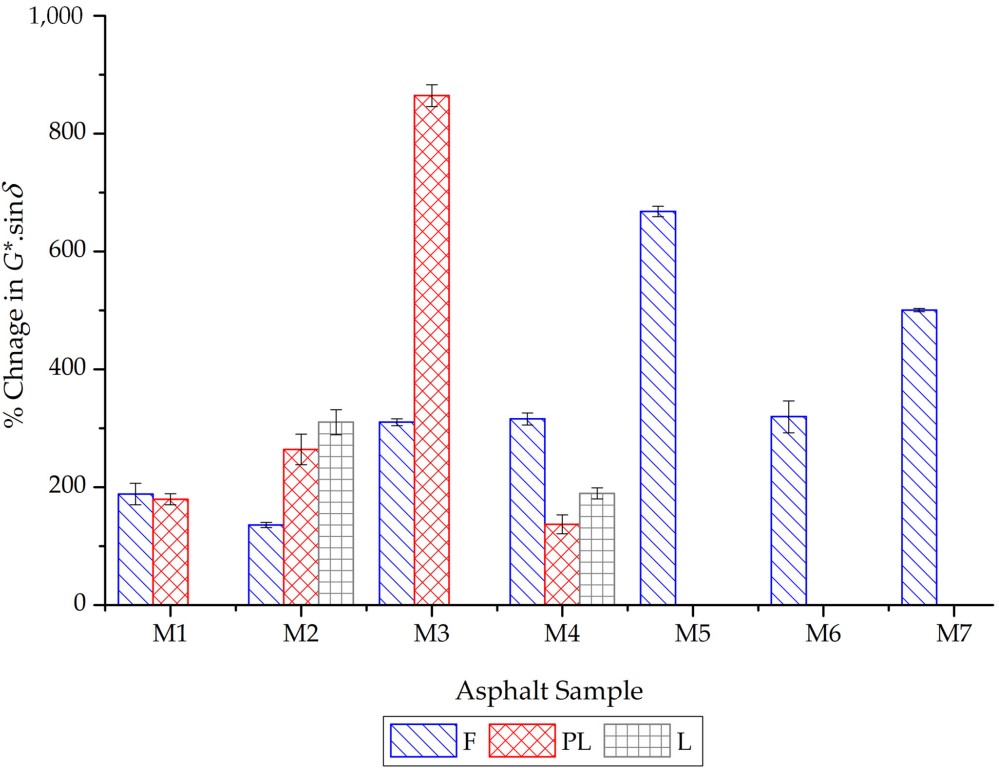

**Figure 8.** Percentage of change in fatigue parameter of the recovered binders.

It is noteworthy that both M2 and M3 contained the same original binder; however, M3 exhibited a 33% ABR by RAS instead of 31% ABR by RAP. The air-blown oxidized binder in the RAS exhibited a detrimental effect on the fatigue and block cracking resistance, as evidenced by the significant disparity in the percentage change in $G^*.\sin\delta$ and G-R between M2 and M3, particularly for PL-recovered binders. This shows that there is a greater effect of the RAS on resistance to fatigue and block cracking because of the extremely oxidized binders. As such, the original and aged binders' compatibility plays an important role in how well the total binder performs. This was evidenced by a 174.39% difference in the percentage change in the $G^*.\sin\delta$ and a 1365.34% difference in the percentage change in the G-R between the M2-F and M3-F. However, this disparity was linked to the PL-recovered binders that saw a 600.53% deviation in percentage change in $G^*.\sin\delta$ and a substantial 255,411.11% jump in percentage change in G-R. Experimental findings indicated that the interaction and compatibility of the RAS binder with the original binder are greater

among PL mixtures than in the F mixtures [12,16]. In practical applications, mixtures with a high ABR%, particularly those incorporating RAS, must be meticulously tailored to achieve a balance between stiffness and ductility. This can be achieved by an adjustment or rejuvenation process of the binder that minimizes excessive aging.

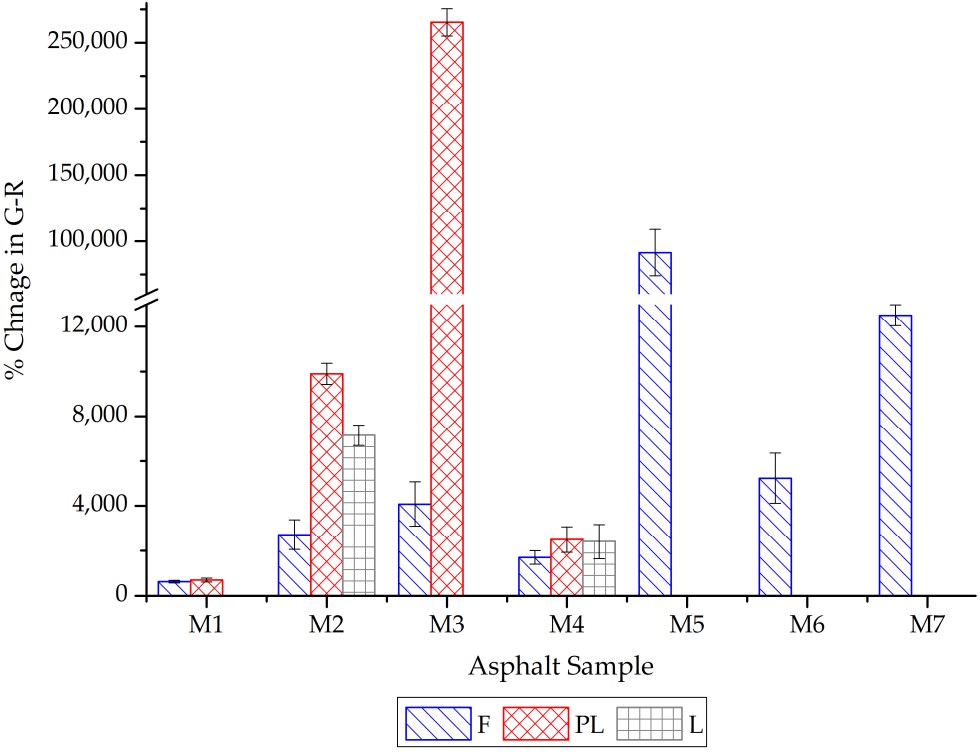

**Figure 9.** Percentage of change in G-R parameter of the recovered binders.

### 2.3.3. Low-Temperature Rheological Analyses

The low-temperature rheological characteristics—negative continuous grade temperature ($T_c$) and delta in negative continuous temperature ($\Delta T_c$)—of recovered and original binders reflect binder aging, as shown in Figure 10. Lower $\Delta T_c$ and higher $T_c$ values depict a decrease in the binder's flexibility that hinders its ability to resist low-temperature cracking. The higher the positive $\Delta T_c$ and the lower the negative $T_c$ values, the better the ability of the binder to release stress due to its good flexibility [40]. Negative $\Delta T_c$ binders tend to exhibit solid behavior rather than liquid at lower temperature ranges, thereby cracking under stress. The negative continuous temperatures were −21.20 °C for the M4-F binder, −19.49 °C for the M4-PL binder, and −19.66 °C for the M4-L binder. Additionally, negative $\Delta T_c$ values, −0.10 °C, −0.51 °C, and −0.46 °C, were found in the M4-F, M4-PL, and M4-L binders, respectively. On the other hand, the −34.69 °C $T_c$ value and positive 0.76 °C $\Delta T_c$ value showed that the M4-O was flexible and had low-temperature crack resistance. The M3, M5, and M6 field mixtures employed the same original binder. The M3, M5, and M6 included 33% ABR: The M5 had 33% ABR by RAP, the M3 had 33% ABR by RAS, and the M6 had 18−15% ABR by RAP-RAS. The recovered binders from these mixtures showed different resistances to low temperatures. The M5-F recovered binder exhibited higher $T_c$ and lower $\Delta T_c$ values compared to the values of the M3-F and the M6-F binders. This occurred due to the RAP and original binders' rapid interactions compared to the RAS and original binders' interactions, which was elucidated in the high- and intermediate-temperature performance results. The results showed that recycled materials influence making binders less flexible and more elastic and solid-like. They were more susceptible to cracking at low temperatures because of this influence.

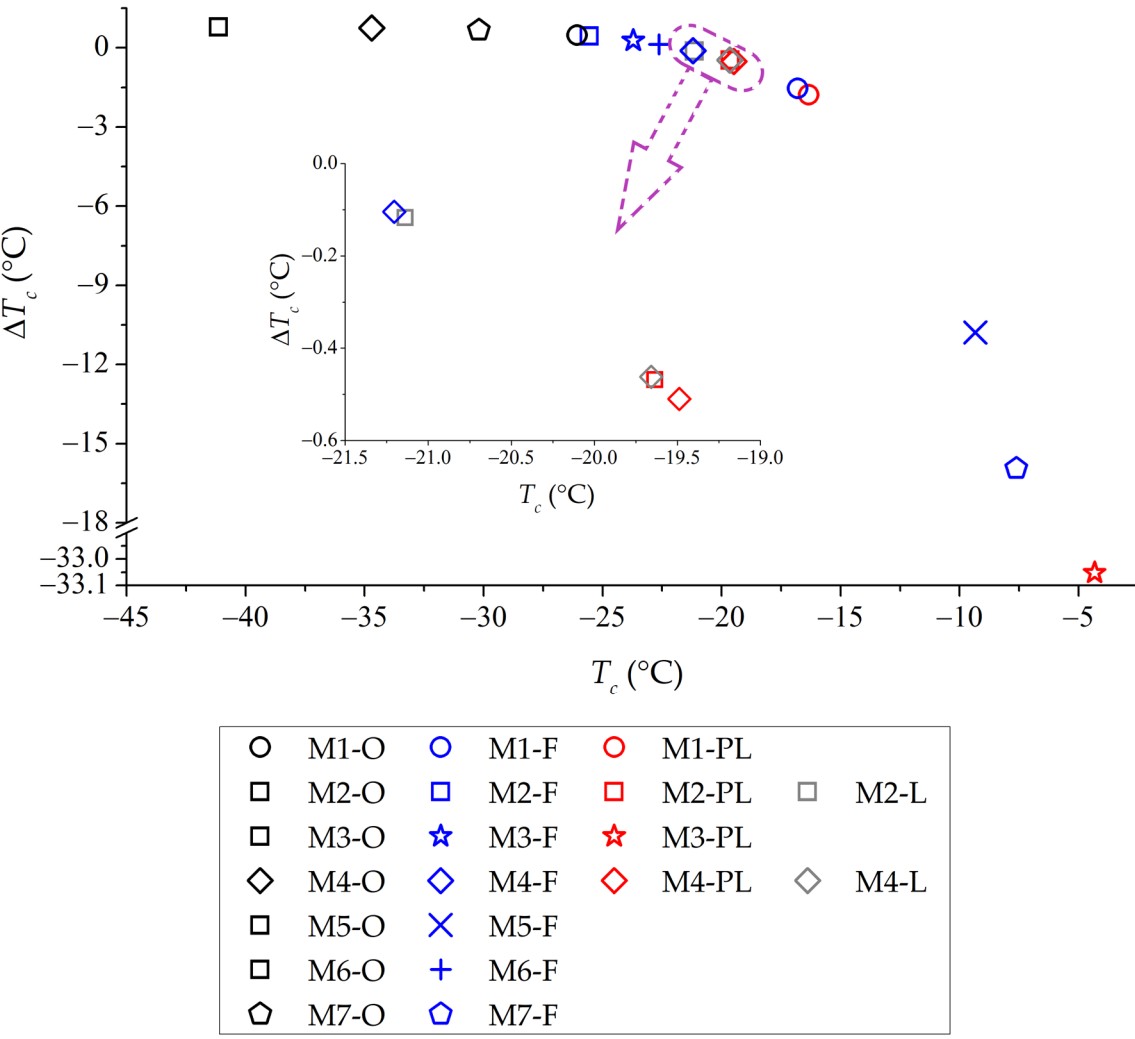

**Figure 10.** Low-temperature properties of original and recovered binders; the purple dashed lines zoom into an area where binders display analogous low-temperature characteristics.

M2 and M3 shared the same original binders, which demonstrated a positive $\Delta T_c$ of 0.79 °C, while the original binder for M4 had the same grade (PG 58–28) and a similar positive $\Delta T_c$ value of 0.76 °C. Nevertheless, increasing the ABR% by RAP from 31% in the M2 to 35% in the M4 led to a progressive increase in $T_c$ and a decline of $\Delta T_c$, deteriorating low-temperature cracking resistance. At the same time, $\Delta T_c$ was shifted from 0.79 °C to −0.47 °C. In M3, whereas the original binder, just like the M2-O, possessed a $T_c$ of −41.13 °C and a positive $\Delta T_c$ (0.79 °C), the PL-recovered $T_c$ of the binder rose significantly to −4.33 °C, and its $\Delta T_c$ fell to −33.05 °C. This is a definite indication of extensive embrittlement; this was noted during the testing program. Figure 11 shows the comparison of the M3-F and M3-PL samples after room temperature stretching: M3-F was flexible, whereas M3-PL was very brittle. In M4, the initial binder with a $T_c$ value of −34.69 °C and a positive $\Delta T_c$ value of 0.76 °C evolved to a $T_c$ of −19.49 °C and a $\Delta T_c$ of −0.51 °C in the PL-recovered binder. These findings provide clear evidence that increasing ABR% results in a steady increase in $T_c$, indicative of higher stiffness, and a decline in $\Delta T_c$, suggesting a reduction in the binder's capacity to release stress and, consequently, an increased propensity for low-temperature cracking.

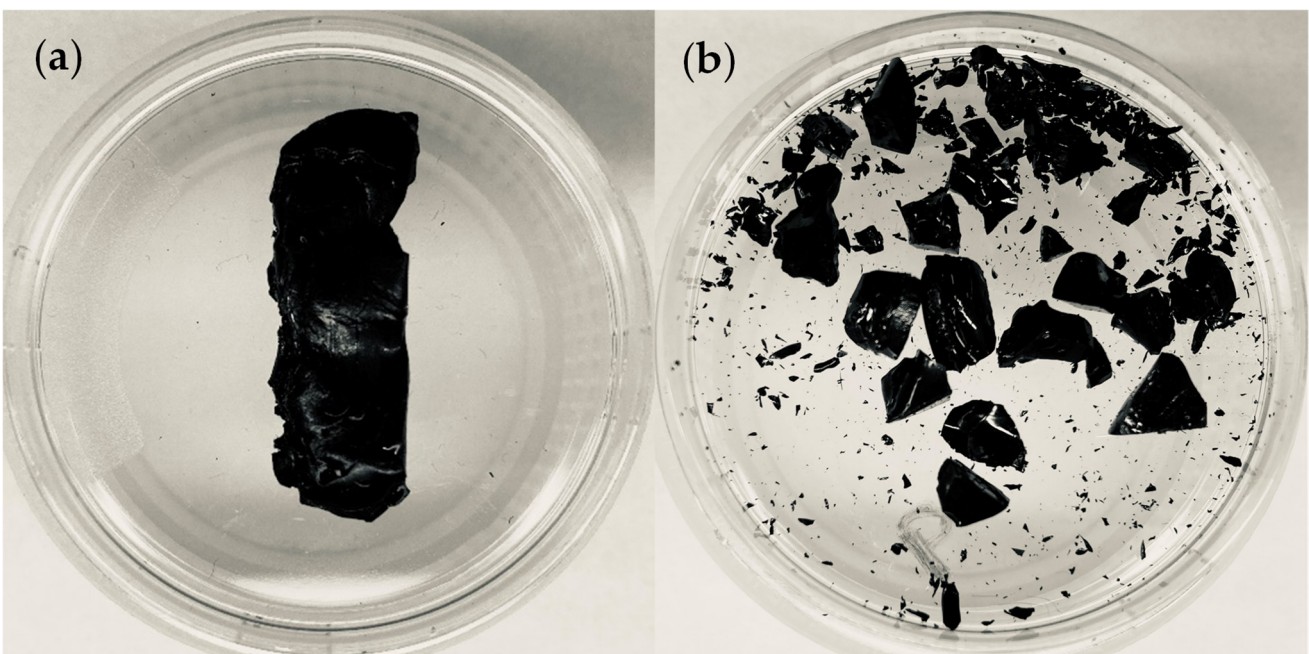

**Figure 11.** M3 recovered binders from (**a**) F; (**b**) PL mixtures.

The mechanism for producing the mixtures indicated that binders obtained from PL mixtures had the least ability to resist cracking at low temperatures, followed by those recovered from L, and finally by F-recovered binders. This conforms to the findings for high-temperature rheological results which showed that both L- and PL-recovered binders exhibited higher stiffness compared to the F-recovered binders. The plant mixtures were reheated in the laboratory at a temperature of $100 \pm 5\,°C$. Then, they were divided into pans before being compacted at the compaction temperature in the JMF. Apart from that, the laboratory-made mixtures were also subjected to short-term aging in the lab and then compacted. These fabrication mechanisms had a significant impact on the interaction processes of the RAP-RAS aged and original binders in PL and L mixtures compared to those observed in the F mixtures.

### 2.3.4. Thermal Analyses

Thermal analysis is a practical approach for assessing recovered binders when only limited amounts are available. Thus, this section intends to assess the outcome of using the thermogravimetric analysis (TGA) in evaluating changes in the various fabricated mixtures (F, PL, and L), which contain RAP-RAS. In turn, the derivative of thermograph (DTG) areas of the original and the recovered binders are shown in Figure 12: The original binders gave the highest areas. The higher the DTG area, the less advanced the aging condition the binder is in, compared to binders having a lower DTG area for which the aging condition is indicated to be elevated [29]. Mixing recycled materials into asphalt mixtures altered the composition of the recovered binders, resulting in higher aging effects and lower DTG areas. For more than 65% of the recovered binders, PL- and L-recovered binders showed lower DTG areas compared with F-recovered binders, which indicated a much stiffer nature. These results agree with the rheological results, which showed that higher stiffnesses could be observed for the PL- and L-recovered binders compared to F mixtures. This was attributed to increased interactions between the aged binders in recycled constituents with those from original binders that occurred in PL and L compared to F mixtures.

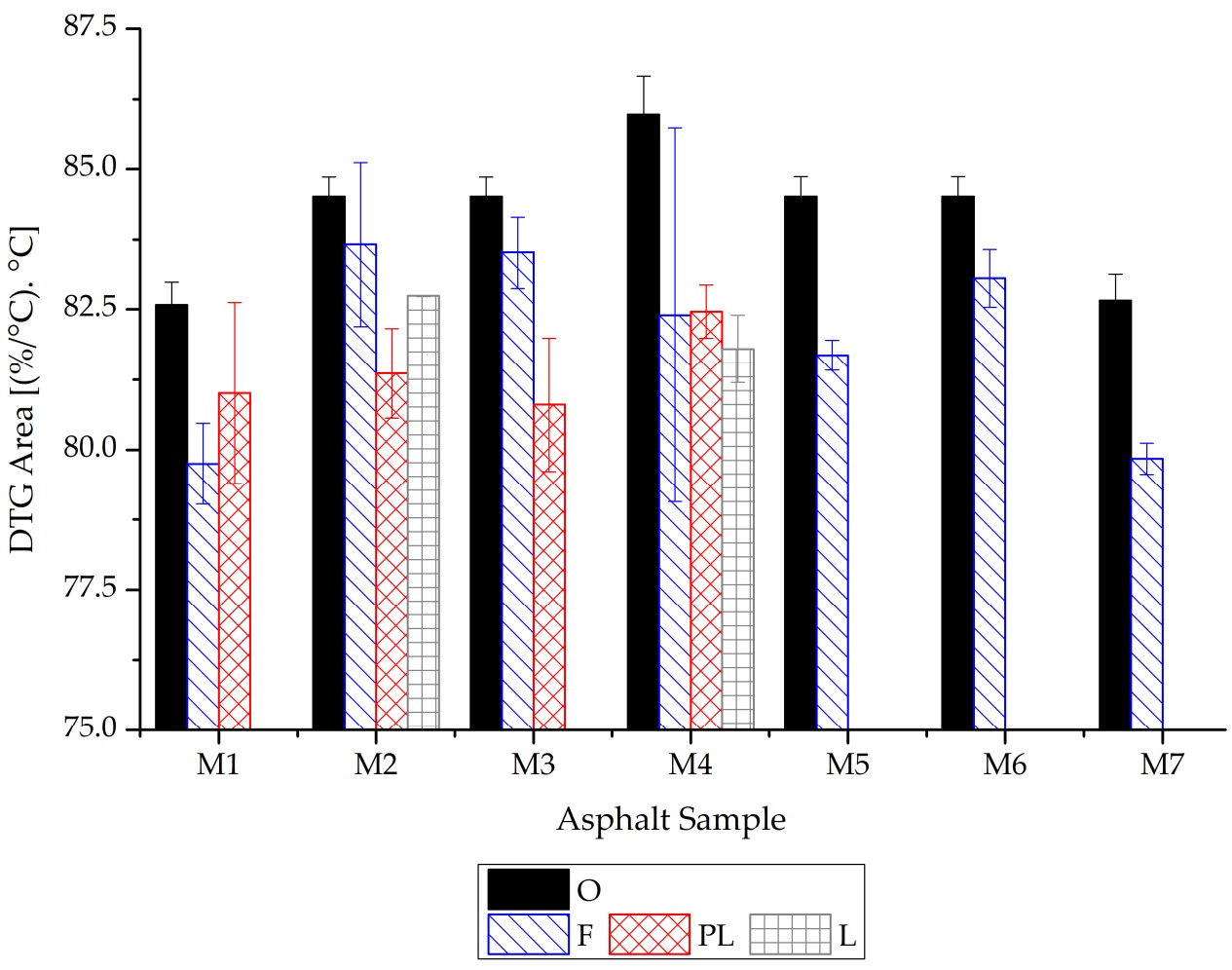

**Figure 12.** DTG areas of original and recovered binders.

A correlation was established between the thermal results, the DTG area, and the rheological properties to facilitate the interpretation of binders' alternations (see Figure 13). Outliers are depicted in a faded gray color. Figure 13a shows a correlation between the DTG area and $G^*/\sin\delta$, which follows an inverse power function. Figure 13b shows an inverse exponential correlation between the DTG area and $G^*.\sin\delta$, while Figure 13c depicts an inverse exponential correlation between the DTG area and the G-R parameter. Figure 13d presents a direct polynomial correlation between the DTG area and $\Delta T_c$. Overall, correlating thermal and rheological properties led to the fact that augmenting the aging components of the recovered binder minimizes the DTG area, thereby increasing the stiffness level. Thus, these outcomes cause higher rutting resistance, higher $G^*/\sin\delta$, lower fatigue and block cracking resistance, higher $G^*.\sin\delta$ and G-R, and lower thermal cracking resistance, as indicated by lower $\Delta T_c$ values.

Figure 14 illustrates the validation of the models presented in Figure 13 through a comparison of measured and predicted thermal characteristics. To enhance the generalizability and robustness of these models, they were validated on the original samples used to create these models and on additional samples. The validation statistical analyses revealed that the standard error per standard deviation of the y-variable (Se/Sy) values were below 0.40 and ranged from 0.22 to 0.39, the coefficient of determination ($R^2$) values ranged from 0.85 to 0.95 (greater than 0.80), and the correlation coefficient (R) values from 0.92 to 0.97 (close to 1.00). These statistical data showed that the models in Figure 13 were reliable when generating predictions.

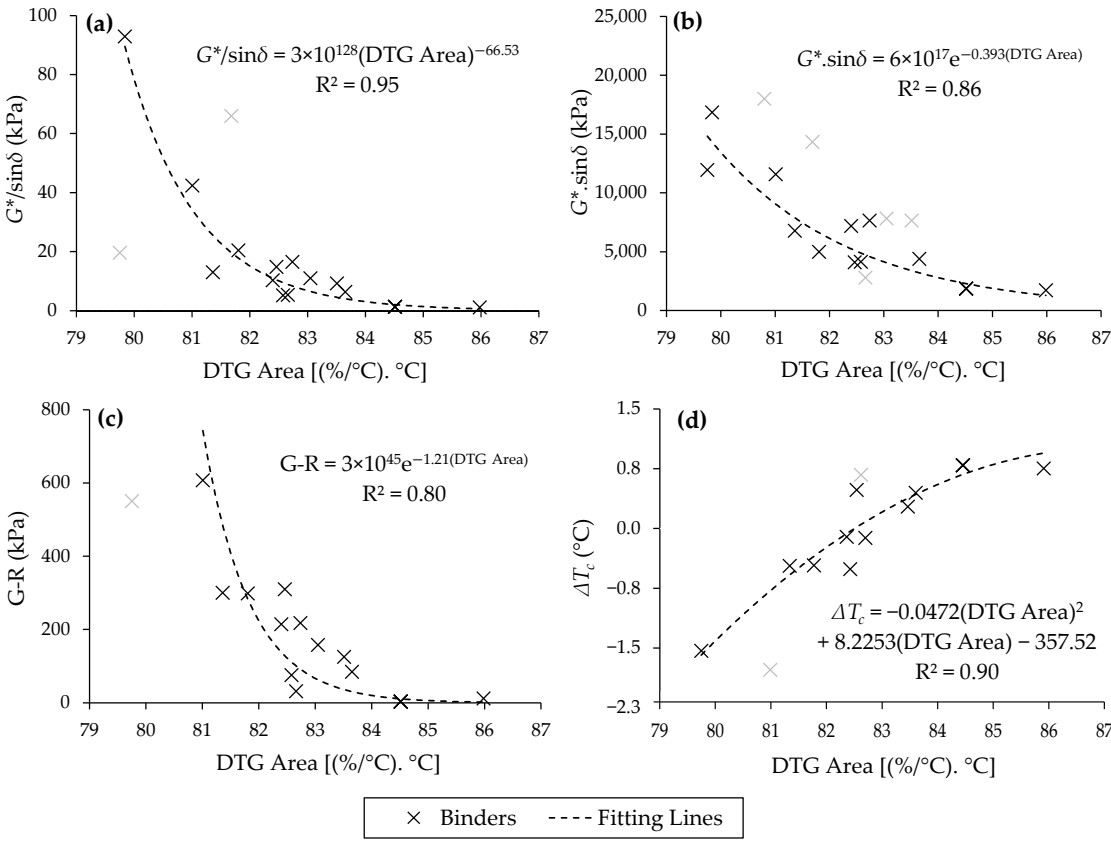

**Figure 13.** Changes in DTG areas versus (**a**) $G^*/\sin\delta$, (**b**) $G^*.\sin\delta$, (**c**) G-R, and (**d**) $\Delta T_c$ of original and recovered binders.

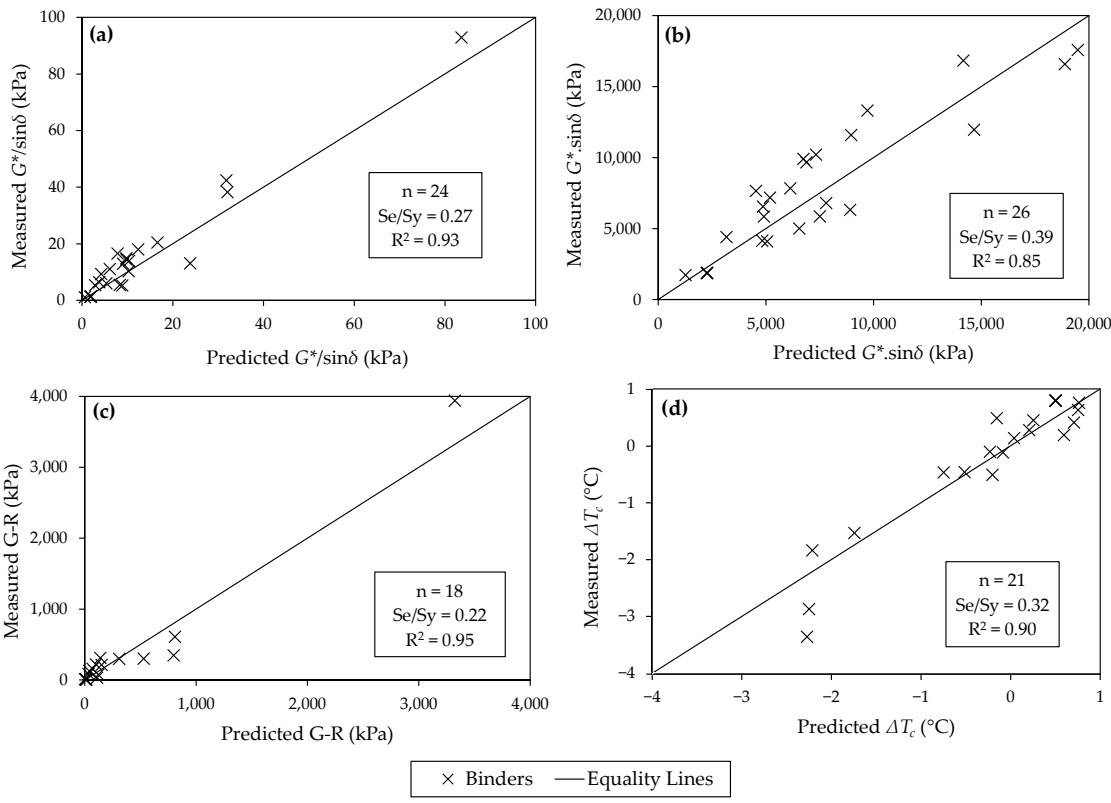

**Figure 14.** Measured vs. predicted rheological analysis: (**a**) $G^*/\sin\delta$; (**b**) $G^*.\sin\delta$; (**c**) G-R; (**d**) $\Delta T_c$.

### 2.4. QA of Mixtures Based on the QA Decision Matrix

Figure 15 displays a QA decision matrix that determines whether the components of the mixtures and the recovered binders comply with the specifications outlined in the JMF and data reported during the QC, with compliance indicated in green and non-compliance in orange. According to the ANOVA, the whole mixtures showed no significant difference between the extracted and designed aggregate gradations and between extracted and contract AC percentages. Based on the $G^*/\sin\delta$ values, the high PG temperatures of the recovered binders were compared to the high PG temperatures of the contract grades. Recovered binders failed by showing higher PG temperatures (stiffer binders) when compared to the contract high PG temperatures except for the M2-F, M3-F, M4-F, and M6-F binders. The low PG temperatures of the recovered and contract binders were compared using the same concept, based on the analyses of the $T_c$ values. Furthermore, the intermediate temperatures of the recovered and contract binders were compared through the analyses of the $G^*.\sin\delta$. All recovered binders had higher intermediate and low PG temperatures (stiffer binders) than the contract binders' values except for M2-F, M3-F, and M6-F binders. Recovered binders with G-R parameter values greater than 180 kPa were in the damage or block cracking zones and thus considered to fail in the QA decision matrix. According to these criteria, all recovered binders were unsuccessful except for the M2-F, M3-F, and M6-F binders. Referring to the AASHTO PP 78 [41], the binders must possess a minimum $\Delta T_c$ value of $-5\,°C$. The more negative $\Delta T_c$, the stiffer the asphalt binder is, and this reflects the inability to relax stress efficiently. Thus, all recovered binders passed except for M3-PL, M5-F, and M7-f. Based on the QA matrix, the overall decision was deemed "Pass" if both the binder and mixture components successfully met all analyses; otherwise, the mixture was classified as "Fail".

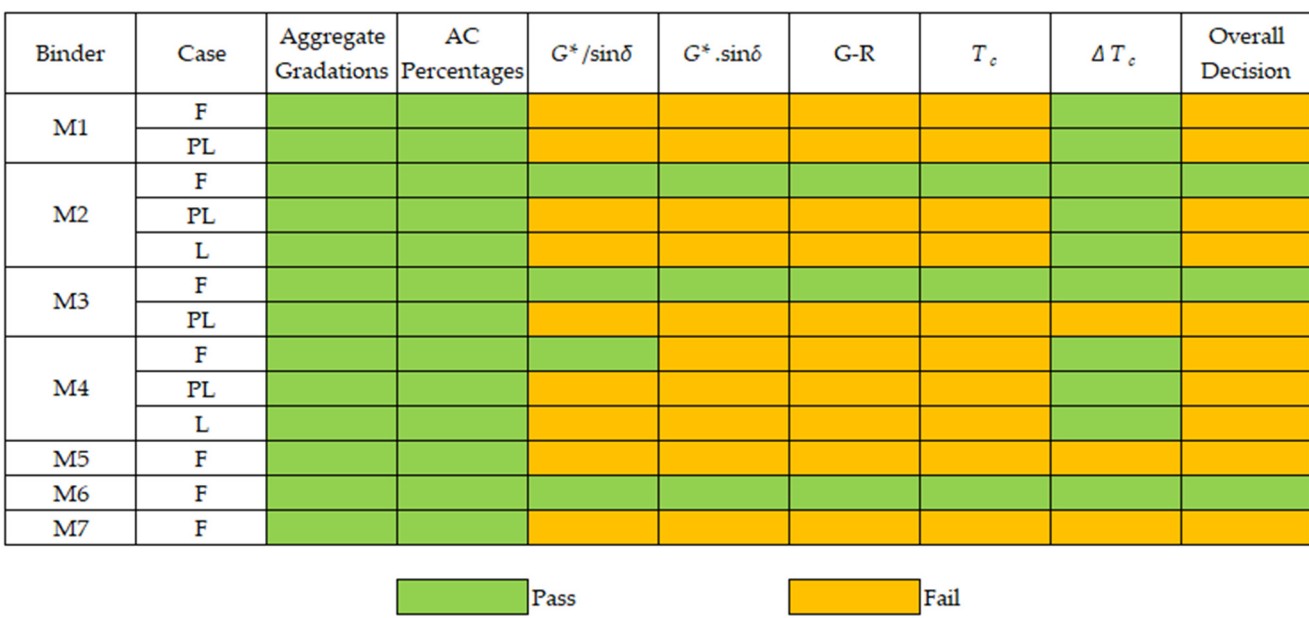

**Figure 15.** QA decision matrix.

In summary, the QA decision matrix offered a complete system to evaluate if recovered binders and mixture constituents satisfy the JMF requirements and the reported QC data. Through the combination of various ANOVA and rheological parameters, the matrix detected the potential for binders' stiffness, aging, and cracking susceptibility problems, interaction mechanisms, and finally, it guided the choice of durable asphalt mixtures. According to the QA decision matrix, approximately 23% of the mixtures complied with the JMF specifications. These mixtures were M2, M3, and M6 field cores with soft binders

(PG 58–28), and two of the mixtures contained RAS (M3 and M6). This highlighted the fabrication mechanism in F-fabricated mixtures that led to the incomplete blending "compatibility" between RAS and original binders. Interestingly, the PL- and L-fabricated mixtures demonstrated that enhanced interactions and compatibility between recycled and original binders accelerated aging and boosted stiffness in the recovered binders. Such intensified interactions, while seemingly beneficial to be understood, could initiate long-term durability concerns, such as reduced stress relaxation capacity and increased susceptibility to cracking.

## 3. Material and Methods

### 3.1. Materials

The data for seven asphalt mixtures were collected, each incorporating varying ABR percentages from RAP-RAS, distinct total AC percentages, original asphalt binders with different PGs, and contract PGs (see Table 5). For the M1 to M4 mixtures, three plant-produced samples were obtained during production from the asphalt plant as loose mixtures, which were preheated and compacted in the lab [plant–lab-compacted (PL) mixtures] to simulate field cores. Additionally, for M1 to M7 mixtures, three field cores (F) per mixture were extracted two weeks post-construction. For mixtures M2 and M4, three lab-simulated mixtures (L) were fabricated and compacted in the laboratory.

**Table 5.** Details of asphalt mixtures.

| Mixture | ABR% by (RAP-RAS) | Total AC% | Original Binder's PG | Contract PG | Number of Samples Collected/Fabricated | | |
|---|---|---|---|---|---|---|---|
| | | | | | F | PL | L |
| M1 | 17% (RAP) | 5.7% | 64H–22 | 70–22 | 3 | 3 | - |
| M2 | 31% (RAP) | 5.1% | 58–28 [a] | 70–22 | 3 | 3 | 3 |
| M3 | 33% (RAS) | 5.2% | 58–28 [a] | 70–22 | 3 | 3 | - |
| M4 | 35% (RAP) | 5.1% | 58–28 | 70–22 | 3 | 3 | 3 |
| M5 | 33% (RAP) | 5.3% | 58–28 [a] | 70–22 | 3 | - | - |
| M6 | 18% (RAP) and 15% (RAS) | 5.2% | 58–28 [a] | 70–22 | 3 | - | - |
| M7 | 35% (RAP) | 4.8% | 64H–22 | 76–22 | 3 | - | - |

[a] Same original binder.

Plant mixtures were collected, preheated at $100 \pm 5$ °C, and compacted in the laboratory at the temperature set in the JMF to simulate the field cores and to assure the quality during the production process. Field mixtures were cored to validate the quality of the mixtures' components after construction. In addition, the lab-simulated mixtures were fabricated to allow an understanding of how the two fabrication mechanisms differ in reality versus in the lab. Quantitative and qualitative analyses of the mixtures' extracted components were carried out to assess the actual values against the contract values as set forth in the JMF. In addition, the results of two plant mixtures without recycled materials from a previous study [37] were discussed to highlight the reverse effect of recycled materials and to clarify the need for a QA framework for mixtures containing recycled materials.

### 3.2. Methods

Asphalt binders were extracted and recovered from each sample, whether cored, collected and compacted, or fabricated from the mixtures presented in Table 5; these are subsequently designated as the "recovered binders". After binder extraction was completed, aggregates were graded and checked carefully against each other (F, PL, and L)

and the designed aggregate gradation in the JMF via ANOVA. Using ANOVA, the contract AC percentages described in the JMF were compared to the extracted AC percentages from several mixes—F, PL, and L. The recovered binders were subjected to comprehensive rheological and thermal evaluations to assess their properties against those specified in the JMF. The rheological analyses included measurements of the $G^*/\sin\delta$ at high temperatures, evaluations of $G^*.\sin\delta$, and G-R parameters at intermediate temperatures, and analyses of the $T_c$ and $\Delta T_c$ values at low temperatures. The analyses of binders and aggregates were ultimately compared to the JMF requirements and the reported QC data utilizing the QA decision matrix. The comprehensive QA framework informing this research is shown in Figure 16. The procedural steps and methodologies are discussed in detail in the following sections for clarity.

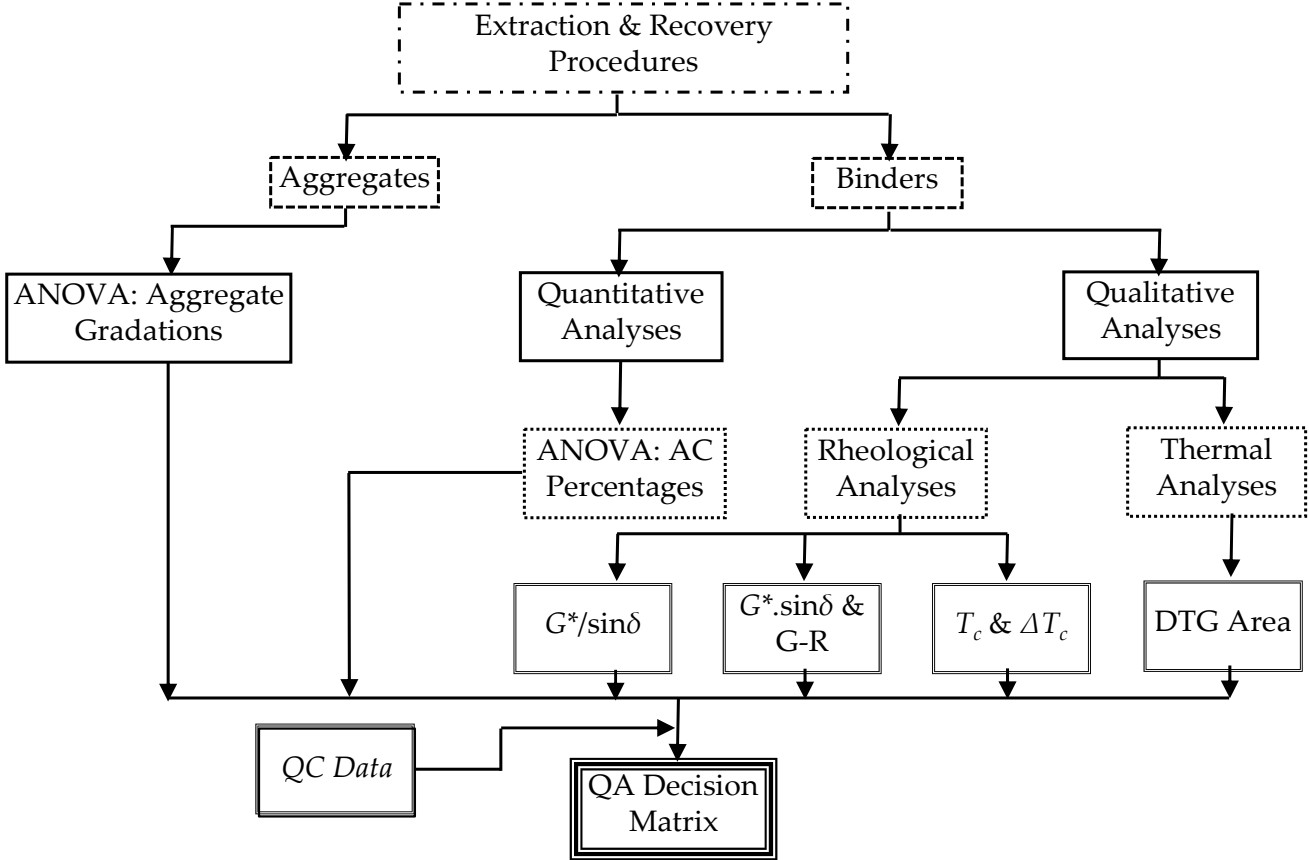

**Figure 16.** QA framework.

### 3.2.1. Extraction and Recovery Procedures

The asphalt mixtures were subjected to heating in an oven at $100 \pm 5\ °C$ for 2 h to promote the breakdown of their components before the extraction and recovery operations. Then, the aggregates and binders were separated using a centrifugal extractor. The binders were recovered from the extracted solvent, after mineral matter removal, using a rotary evaporator.

#### Aggregates and Binders Extraction from Asphalt Mixtures

Asphalt binders were extracted from mixtures by a centrifugal extractor as per ASTM D2172/D2172M-24 [42]. An amount of 1200 mL of trichloroethylene (TCE) was utilized for wet soaking the asphalt mixtures for 55 min and extracting the asphalt binders. Then, the centrifuge started, and the speed gradually increased step by step until the effluent had stopped. Next, 200 mL of TCE was poured into the contents of the centrifuge bowl and

allowed to stand for 5 min before restarting the centrifuge. The procedure was performed repeatedly until a straw color effluent was reached. Next, 200 mL of extracted effluent was poured into two ignition dishes and placed in the ignition oven at 600 °C for one and a half hours to determine the mineral matter in the extracted effluent (ashing mineral matter calculation). The remaining effluent mineral matter was removed and calculated using a filterless centrifuge extractor (centrifuge mineral matter calculation).

Recovery of Binders from Binder-TCE Solvents

The asphalt binders were recovered from the extracted effluent via a rotary evaporator, following ASTM D5404/D5404M-24 [43], after the mineral matter removal. The rotary evaporation procedure consisted of evaporating the TCE from a flask that held 600 mL of the binder dissolved in TCE under a 40 mm Hg vacuum below atmospheric pressure. At first, the flask was dipped into an oil bath boiling at 140 °C; the initial depth of immersion was set to control the evaporation rate of the solvent. During the process, the flask was spun at 40 revolutions per minute. There was an inlet to a continuous source of carbon dioxide gas at 500 mL/min to establish and keep up with the inert environment, thus preventing oxidation. Following the distillation of the majority of the solvent and the observation of minor condensation in the condenser, the procedure was intensified for the final 10 min. During this phase, the flask was immersed in water 40 mm deep, the vacuum was increased to 600 mm Hg below atmospheric pressure, the flow of carbon dioxide was increased to 600 mL/min, and the rotational speed of the flask was increased to 45 rpm. Following this time, the flask was turned over and set in the oven at 165 °C for a further 10 min to facilitate the passage of the asphalt binder into a can.

### 3.2.2. Aggregate Gradations

Following binder extraction, the aggregates were dried for three hours in an oven at $110 \pm 5$ °C to remove residual traces of TCE. Aggregate gradations were performed on these aggregates after cooling to room temperature and compared with each other (F, PL, and L) and with the JMF gradations by ANOVA. The aggregate gradations were analyzed using sieves ranging from 2 inches to No. 200.

### 3.2.3. Analyses of Binders

Following the extraction and recovery procedures, the binders were evaluated both quantitatively and qualitatively as explained in the subsequent sections.

Quantitative Analyses

The AC percentage was calculated according to the mineral matter calculation method. The total AC percentage in Table 5 was compared with the extracted AC percentage using centrifuge and ashing mineral matter determination methods via ANOVA.

Qualitative Analyses

This section outlines the procedures undertaken to perform the rheological and thermal investigations of the original and recovered binders.

1. Aging of Original Binders

Original asphalt binders have undergone aging in accordance with ASTM D2872-22 [44] using a rolling thin-film oven (RTFO). A 35 g sample of each asphalt binder was placed in an RTFO bottle and heated at 163 °C with a flow rate of $400 \pm 25$ mL/min for 85 min.

2. Rheological Analyses

The rheological properties of the recovered and RTFO-aged original binders were assessed using a dynamic shear rheometer in accordance with ASTM D7175-15 [45]. To ensure the reproducibility of the results, each binder was tested using two replicate samples, and the average results were assessed. The $G^*/\sin\delta$ rutting parameter was examined for binder samples with a thickness of 1 mm and a diameter of 25 mm at a temperature range starting with 52 °C and ending with 76 °C (or higher based on the binder stiffness results), a frequency of 10 rad/s, and a 10% shear strain. The fatigue resistance of the binders was tested for 8 mm diameter and 2 mm thickness specimens at 22 °C till 34 °C, a 10 rad/s frequency, and 1% shear strain by measuring the $G^*.\sin\delta$ as a fatigue parameter. In addition, the G-R parameter was evaluated and calculated via Equation (1) at 15 °C, with a frequency of 0.005 rad/s and a 1% shear strain [10,39,46]. The thermal cracking analyses of the binders were analyzed by predicting the $T_c$ and $\Delta T_c$ using Equations (2) and (3) [10], respectively.

The G-R parameter was calculated by applying Equation (1) as follows:

$$\text{G} - \text{R} = G * \left( \frac{(\cos \delta)^2}{\sin \delta} \right), \tag{1}$$

where G-R is the Glover–Rowe parameter in kPa, $G^*$ is the complex shear modulus in kPa, and $\delta$ is the phase angle in degrees.

$T_c$ was predicted using Equation (2), which is as follows:

$$T_c = \frac{\ln\left[ \frac{(\text{G}-\text{R})}{20128} \right]}{0.2142}, \tag{2}$$

where $T_c$ is the negative continuous grade temperature (°C) and G-R is the Glover–Rowe parameter in kPa.

Using Equation (3), $\Delta T_c$ was determined as follows:

$$\Delta T_c = \frac{(\text{G} - \text{R}) - 189.82}{-235.26} \tag{3}$$

where $\Delta T_c$ is the delta in negative continuous temperature (°C), and G-R is the Glover–Rowe parameter in kPa.

3. Thermal Analyses

The asphalt binders' thermal behaviors were assessed using TGA following ASTM E1131-20 [47]. Small binder samples (20–25 mg) were heated from room temperature to 600 °C at 50 °C/min under 60 mL/min of nitrogen flow. The study used the high-resolution dynamic method patented by TA® Instruments. By changing the temperature in real time as the sample loses mass, this method improves the measurement accuracy and reliability [48]. Unlike conventional constant temperature heating, the technology automatically decreases the temperature if the sample loses weight too fast, so it improves the resolution. This method increases test efficiency and measurement accuracy [49]. The DTG areas for the tested binders were compared to better understand the binders' thermal decompositions and how this influences the performance at different temperatures.

3.2.4. QA Decision Matrix

The QA decision matrix was created to assist decision makers in accurately assessing the contractor's performance and identifying the compliance of the materials to JMF requirements. This matrix involved statistical comparisons and performance-based evaluations. The designated targets in the JMF (e.g., aggregate gradations, asphalt contents, and contract binder performance parameters) were compared with the corresponding values

for the components extracted and recovered from each mixture. Finally, the overall compliance was determined for each mixture, facilitating differentiation among the different fabricated mixtures.

## 4. Conclusions

The high variability of recycled materials necessitates an enhanced QA framework to ensure compliance of the materials with the JMF and QC data, resulting in durable pavements. The fabrication mechanism was among various influencing factors regulating the interactions between RAP-RAS and original binders, influencing the overall recovered binder's performance. The limited amounts of the recovered binders were another concern, especially from field cores. Thus, a thermal analysis was introduced as an effective and reliable technique to predict the binder's performance via thermal-rheological models. Statistical analyses and performance evaluation parameters were integrated into a QA decision matrix to introduce a rational tool to monitor and regulate the contractors' efficiency and materials' compliance to the JMF requirements. Based on the recommended QA framework, the following conclusions were drawn:

1.  For mixtures including recycled materials, the proposed QA framework is a necessary process displaying a thorough approach for component evaluation.
2.  The component-level analyses revealed that aggregate sizes and asphalt contents showed compliance with the JMF targets, reflecting minimal disruption from the extraction process.
3.  The recovered binders, particularly PL and L binders, had significant alterations when compared to the contract binders in the JMF, with increased stiffnesses, greater elasticity values, and reduced capabilities to relax thermal stresses, reflecting the need for QA framework targeting the behavior of the recovered binders.
4.  Increasing the ABR% by RAP-RAS enhanced the rutting resistance; however, heightened susceptibility to fatigue, block, and thermal cracking, confirmed the need for a QA decision matrix to balance the performance evaluation.
5.  RAP binders interacted readily with the original binders; however, the RAS binders had a delayed interaction, especially in F mixtures. This reflected the need for a QA decision matrix to compare the recovered binders from different fabricated mixtures.
6.  The QA decision matrix is an effective approach for assessing and determining the adherence of asphalt mixtures to JMF specifications, combining statistical analysis with performance-based evaluation.

## 5. Recommendations and Future Work

1.  This study advocates the use of the QA framework and the decision matrix for asphalt mixtures containing recycled materials. These tools will facilitate materials compliance monitoring for decision makers.
2.  It is recommended that thermal analyses and the proposed models be used to assess the performance of the recovered binders in limited quantities.
3.  A more in-depth investigation of the lab's aging and compaction mechanisms, as well as their impact on the interactions between the original and aged binders, is required. Understanding these interactions has the potential to significantly influence the mixtures' long-term performance.

**Author Contributions:** Conceptualization, M.A. and E.D.-A.; methodology, M.A. and E.D.-A.; software, E.D.-A. and M.A.; validation, M.A. and E.D.-A.; formal analysis, E.D.-A. and M.A.; investigation, E.D.-A. and M.A.; resources, M.A.; data curation, M.A. and E.D.-A.; writing—original draft preparation, E.D.-A. and M.A.; writing—review and editing, E.D.-A. and M.A.; visualization, M.A. and

E.D.-A.; supervision, M.A.; project administration, M.A.; funding acquisition, M.A. All authors have read and agreed to the published version of the manuscript.

**Funding:** This research received no external funding.

**Data Availability Statement:** The original contributions presented in this study are included in the article. Further inquiries can be directed to the corresponding author(s).

**Acknowledgments:** The authors thank MoDOT for supplying part of the asphalt mixtures (project number TR201807).

**Conflicts of Interest:** The authors declare no conflicts of interest.

## Abbreviations

The following abbreviations are used in this manuscript:

| | |
|---|---|
| ABR | Asphalt Binder Replacement |
| AC | Asphalt Content |
| ANOVA | Analysis of Variance |
| DTG | Derivative of Thermograph |
| F | Field |
| G-R | Glover–Rowe |
| JMF | Job Mix Formula |
| P | Plant |
| PG | Performance Grade |
| PL | Plant and Lab Compacted |
| QA | Quality Assurance |
| QC | Quality Control |
| RAS | Recycled Asphalt Shingles |
| RAP | Reclaimed Asphalt Pavement |
| RTFO | Rolling Thin-Film Oven |
| TCE | Trichloroethylene |
| TGA | Thermogravimetric Analysis |

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
