# Peer review of "Quality Assurance Framework for Recovered Binders and Aggregates from Asphalt Mixtures Incorporating Recycled Materials"

_recycling, doi:10.3390/recycling10020071_

Round 1

Reviewer 1 Report

Comments and Suggestions for Authors

This manuscript proposed a quality assurance framework for asphalt mixtures that contained recycled materials, aiming at how to use recycled materials without sacrificing the performance of asphalt mixtures. Its research program was well planned and test results were presented correctly. It adds valuable technical for the recycle asphalt mixture design. Here are some comments for the improvement: 1) Figure 1 was poorly prepared, it need extensively revision. 2) Figure 2 is difficult to read. Cannot they put the curves in one figure, so that their differences can be easily figure out? 3) Most of the data figures were not well designed for scientific research paper. 4) A list of traditional tests was proposed and conducted on recovered binder and aggregate from recycled mixture. There are many literatures reported similar experimental study. What is new in your work? And what may your research different from others.

Author Response

Manuscript ID: recycling-3541203

We sincerely value the thoughtful input and suggestions from the editor and reviewers. We have reviewed their feedback and incorporated relevant points where warranted to improve clarity, accuracy, and quality of the study. The table below contains additional clarifications and responses.

--------------------------------------------------------------------------------------------------------------------

This manuscript proposed a quality assurance framework for asphalt mixtures that contained recycled materials, aiming at how to use recycled materials without sacrificing the performance of asphalt mixtures. Its research program was well planned and test results were presented correctly. It adds valuable technical for the recycle asphalt mixture design. Here are some comments for the improvement:

---------------------------------------------------------------------------------------------------------------------

Comment 1: Figure 1 was poorly prepared, it need extensively revision.

Response 1: Thank you for this valuable observation. Figure 1 was revised and rearranged. The updated figure now better expresses the quality assurance framework and the proposed quality assurance decision matrix. 

-----------------------------------------------------------------------------------------------------------------------

Comment 2: Figure 2 is difficult to read. Cannot they put the curves in one figure, so that their differences can be easily figure out?

Response 2: Thank you for your remark. It was followed and taken into consideration. Based on that, Figure 2 was separated into two subfigures. Figure 2a depicts the aggregate gradations for the JMF and extracted aggregates using No. 16 and larger sieves. Furthermore, Figure 2b shows the gradations of the same aggregates on sieves smaller than No. 16. This improves clarity and facilitates comparisons among the various sieve sizes for the extracted and designed aggregate gradations. 

----------------------------------------------------------------------------------------------------------------------

Comment 3: Most of the data figures were not well designed for scientific research paper.

Response 3: Thank you for your valuable observation, which was applied to all figures and improved the paper's quality. The following modifications were made in the body of the manuscript:

  1. Figures 4 to 13 were formatted and colored according to standard guidelines: The colors black, blue, red, and gray stood for the contract/original, F, PL, and L binders, respectively.
  2. To increase clarity, the horizontal and vertical scales of the figures were altered. Moreover, error bars were added to Figures 9 and 10 to show data variability.
  3. A new batch of asphalt mixtures was examined, including the M5, M6, and M7 mixtures. These samples were used to analyze the changes in the recovered binders from M2, M3, M5, and M6 mixtures that all had the same original binder, while also examining different percentages of ABR% by RAP/RAS. These analyses reflected the effect of the variability of the recycled components on changing the binder’s performance.
  4. The modified figures demonstrated the distinct mechanisms involved in the interaction processes between RAP/RAS and original binders in F, PL, and L mixtures, and illustrated the variability of RAP sources.
  5. In Figure 16, the authors present the quality assurance decision matrix, which provides support for decision-makers and road authorities to manage and control the contractor performance.

-------------------------------------------------------------------------------------------------------------------

Comment 4: A list of traditional tests was proposed and conducted on recovered binder and aggregate from recycled mixture. There are many literatures reported similar experimental study. What is new in your work? And what may your research different from others.

Response 4: The authors appreciate your insightful comment. This was considered in the amended version of the paper. The introduction focused on the study's challenges and novel methods. Previous research investigated the aggregates and binders from a quality control perspective; however, our study focused on a quality assurance framework and decision matrix. The innovative ideas of this study are summarized in the following points:

  1. Quality Assurance Framework and Decision Matrix: This study focused on the quality assurance framework unlike other studies that aimed to qualify the extracted binders and aggregates from the quality control perspective.
  2. Interactions between RAP/RAS and Original Binders: The quality assurance framework was proposed by taking the recycled material’s variability into consideration, as well as the interactions between the RAP/RAS and original binders in different fabricated mixes (field, plant, and lab).
  3. Thermal Analysis: The thermal analysis was a rational tool to predict the performance of the recovered binders with a limited amount through predicted models proposed in this study. The prediction models were verified on the samples used to create the model and other samples to enhance the models’ generalizability and robustly.

Reviewer 2 Report

Comments and Suggestions for Authors

The title and the aim in Line 89 suggest that the focus of the manuscript would be on the quality assurance framework. However, the QA framework in Figure 1 was presented without explanation and justification, and the details of the manuscript are on the comparison of the results of various tests on specimens prepared in three different ways and of four different mixtures. These are not directly related to the “systematic oversight and verification of the contractor's QC procedures to ensure their effectiveness and that the final product meets all contractual and regulatory requirements” as presented in the definition of QA in Lines 44-46. It is suggested to revise the title and the research objective to better match the content of the manuscript, and add discussion in the manuscript on how the test plan, results, and analysis can help QA.

Both the abstract and the conclusions should be revised to be more concise, avoiding including specific numbers (e.g., p-value, F-value) in them.

Error bars are missing in Figures 5 and 6.

The scope of the study is limited, with only four mixtures and three specimen replicates included in the experimental design. Considering the variability of mixture properties in the field pavement due to a variety of reasons, it seems inadequate to take only three specimens from the field.

Author Response

Manuscript ID: recycling-3541203

We sincerely value the thoughtful input and suggestions from the editor and reviewers. We have reviewed their feedback and incorporated relevant points where warranted to improve clarity, accuracy, and quality of the study. The table below contains additional clarifications and responses.

-------------------------------------------------------------------------------------------------------------------

Comment 1: The title and the aim in Line 89 suggest that the focus of the manuscript would be on the quality assurance framework. However, the QA framework in Figure 1 was presented without explanation and justification, and the details of the manuscript are on the comparison of the results of various tests on specimens prepared in three different ways and of four different mixtures. These are not directly related to the “systematic oversight and verification of the contractor's QC procedures to ensure their effectiveness and that the final product meets all contractual and regulatory requirements” as presented in the definition of QA in Lines 44-46. It is suggested to revise the title and the research objective to better match the content of the manuscript, and add discussion in the manuscript on how the test plan, results, and analysis can help QA.

Response 1: Thank you for the insightful feedback. Figure 1 was restructured to more clearly address the quality assurance framework’s elements, culminating in a quality assurance decision matrix. Working with this matrix enables the road authorities and decision makers to oversee and regulate the contractors’ performance based on the quality assurance standpoint after making comparisons with the quality control data. The structure of the test plan and analyses were modified accordingly.

Also, the scope of the study had been reinforced and expanded by including additional field cores (M5, M6, and M7). This enhanced the analysis and captured the variability principle between the recycled components’ sources.

The comparisons of the binders' characteristics aimed to elucidate the principles of the interaction process. These interactions may be different for the same mixtures including the same components but fabricated differently (e.g., M3 mixture).

Thermal analysis, in conjunction with rheological analyses, was crucial for predicting the performance of binders in limited quantities through models validated on both the original samples and additional samples to improve the generalizability and robustness of the models.

--------------------------------------------------------------------------------------------------------------------

Comment 2: Both the abstract and the conclusions should be revised to be more concise, avoiding including specific numbers (e.g., p-value, F-value) in them.

Response 2: I appreciate your thoughtful feedback. The expanded scope of the study and the additional analysis led to revisions in the abstract and conclusions. To improve reading and guarantee clarity, numerical values (such as the p-value and F-value, etc.) were not included in the abstract or conclusions. Particularly for mixtures incorporating recycled components (RAP and/or RAS), the abstract and conclusions stressed the value of the quality assurance framework and decision matrix over the quality control processes. In order to enable the road authority to proactively examine the contractor's efficiency, compliance with the JMF's standards, and reinforcement of the idea of long-term, durable pavements, the quality assurance matrix incorporates statistical analysis of mixture components and performance evaluations for the binders.

--------------------------------------------------------------------------------------------------------------------

Comment 3: Error bars are missing in Figures 5 and 6.

Response 3: Thank you for the valuable note. Error bars were added to Figures 5 and 6. However, in Figure 6, errors bars are not visible for some columns because the corresponding STD values were zero.

-------------------------------------------------------------------------------------------------------------------

Comment 4: The scope of the study is limited, with only four mixtures and three specimen replicates included in the experimental design. Considering the variability of mixture properties in the field pavement due to a variety of reasons, it seems inadequate to take only three specimens from the field.

Response 4: Thank you for your insightful comment. Acquiring a large number of field cores was faced with challenges, including limited material availability and coordination with road agencies that required permissions and times. Moreover, the authors planned to have the plant and field mixes from the same mixes that took time to collect. The number of replicates was likewise affected by these challenges.

In order to address the concerns raised from this point, the authors have heeded your suggestion and have expanded the study to include more core samples (M5, M6, and M7) in the field. Three cores were taken for each mixture within two weeks of being constructed to include more data on field variability; particularly for mixtures M2, M3, M5, and M6 where original binders were matched. Furthermore, the quality assurance matrix proposed in Figure 16, is adaptable to the inclusion of future data sets.

--------------------------------------------------------------------------------------------------------------------

Reviewer 3 Report

Comments and Suggestions for Authors

This manuscript suggested a quality assurance framework for the asphalt mixtures that incorporate recycled materials. The manuscript is written well and can contribute to the asphalt pavement research. Recommendations and comments are listed below for reference.

  1. There is existing literature investigating/documenting the framework of incorporating recycling binder and aggregates in the mixtures. Please include those literatures and explain what is new and significant (difference) in this manuscript?
  2. In Abstract, please include quantitative comparisons (observed percentage changes in terms of QA framework) among the mixtures.

Author Response

Manuscript ID: recycling-3541203

We sincerely value the thoughtful input and suggestions from the editor and reviewers. We have reviewed their feedback and incorporated relevant points where warranted to improve clarity, accuracy, and quality of the study. The table below contains additional clarifications and responses.

------------------------------------------------------------------------------------------------------------------

This manuscript suggested a quality assurance framework for the asphalt mixtures that incorporate recycled materials. The manuscript is written well and can contribute to the asphalt pavement research. Recommendations and comments are listed below for reference.

-------------------------------------------------------------------------------------------------------------------

Comment 1: There is existing literature investigating/documenting the framework of incorporating recycling binder and aggregates in the mixtures. Please include those literatures and explain what is new and significant (difference) in this manuscript?

Response 1: Thank you for your valuable feedback. The introduction was rewritten, and additional references were included. Several investigations examined the extracted components from mixtures incorporating recycled components from a quality control standpoint, and their findings are reported in the body of the article. Nonetheless, the manuscript's fundamental objective is to provide a quality assurance framework that combines statistical analysis and performance testing. These are included in a quality assurance decision matrix as a logical tool for advising decision makers on the overall significant consequences of recycled materials. Unlike previous research that primarily focused on quality control methods or limited quality assurance to components, the quality assurance decision matrix may refine the differences between fabrication mechanisms in different mixes by understanding the overall decision “Pass” or “Fail” and comparing the matrix's sub parameters.

------------------------------------------------------------------------------------------------------------------

Comment 2: In Abstract, please include quantitative comparisons (observed percentage changes in terms of QA framework) among the mixtures.

Response 2: Thank you for your valuable comment that improved the abstract's clarity. Based on this, the abstract was reviewed, organized, as well as rewritten to better reflect the study's overall objective, especially after analyzing the new dataset (M5, M6, and M7 field cores).  A key quantitative comparison was included in the abstract focusing on the overall acceptance outcome based upon the quality assurance decision matrix. Specifically, only 23% of the mixtures satisfied the JMF requirements, highlighting the effect of using soft binders and fabrication mechanisms in field mixtures. This level of quantitative analysis reflects the importance of the quality assurance decision matrix as a rational tool for screening the asphalt mixtures including recycled components.

-------------------------------------------------------------------------------------------------------------------

Round 2

Reviewer 3 Report

Comments and Suggestions for Authors

The authors addressed the reviewer comments.